# Transcriptional regulators ensuring specific gene expression and decision-making at high TGFβ doses

Laura Hartmann[1,2] , Panajot Kristofori[1,2] , Congxin Li[1,2] , Kolja Becker[1], Lorenz Hexemer[1,2], Stefan Bohn[3], Sonja Lenhardt[3], Sylvia Weiss[1,2], Björn Voss[4], Alexander Loewer[3] , Stefan Legewie[1,2]

**TGFβ-signaling regulates cancer progression by controlling cell division, migration, and death. These outcomes are mediated by gene expression changes, but the mechanisms of decision-making toward specific fates remain unclear. Here, we combine SMAD transcription factor imaging, genome-wide RNA sequencing, and morphological assays to quantitatively link signaling, gene expression, and fate decisions in mammary epithelial cells. Fitting genome-wide kinetic models to our time-resolved data, we find that most of the TGFβ target genes can be explained as direct targets of SMAD transcription factors, whereas the remainder show signs of complex regulation, involving delayed regulation and strong amplification at high TGFβ doses. Knock-down experiments followed by global RNA sequencing revealed transcription factors interacting with SMADs in feedforward loops to control delayed and dose-discriminating target genes, thereby reinforcing the specific epithelial-to-mesenchymal transition at high TGFβ doses. We identified early repressors, preventing premature activation, and a late activator, boosting gene expression responses for a sufficiently strong TGFβ stimulus. Taken together, we present a global view of TGFβ-dependent gene regulation and describe specificity mechanisms reinforcing cellular decision-making.**

## Introduction

The TGFβ-signaling pathway controls embryonic development and adult tissue homeostasis. Its dysregulation has been linked to various diseases such as fibrosis and cancer (Akhurst & Hata, 2012). During cancer progression, TGFβ plays a dual role. In early stages of tumorigenesis, the TGFβ-signaling pathway has tumor-suppressive properties by promoting apoptosis and inhibiting cell proliferation. In later stages, it acts as a tumor promoter by preventing cell cycle arrest and initiating epithelial-mesenchymal transition (EMT), which is a hallmark of cancer, to acquire migratory properties by down-regulation of epithelial and up-regulation of mesenchymal markers (Siegel & Massagué, 2003; Xu et al, 2009; Ikushima & Miyazono, 2010). The underlying molecular mechanisms controlling the shift from tumor suppressor to tumor promoter function are currently unclear but may involve differential expression of transcriptional factors (TFs) and alterations in the temporal dynamics of signaling (Piek et al, 2001; Mullen et al, 2011).

TGFβ functions as an extracellular cytokine that triggers intracellular signaling by binding to and activating its transmembrane serine/threonine kinase receptors, TGFβRI and TGFβRII. The activated receptors phosphorylate cytoplasmic signal transducers, SMAD2, and SMAD3, which subsequently heterotrimerize with SMAD4, translocate to the nucleus, and function as TFs by binding to the promoters of target genes (Feng & Derynck, 2005). The SMAD2/3/4 complex binding has low affinity to SMAD-binding elements in promoter regions and transcriptional regulation by SMAD is therefore dependent on additional proteins such as TFs and epigenetic regulators (Shi et al, 1998; Morikawa et al, 2013; Hill, 2016). SMADs use their Mad Homology 1 (MH1) and Mad Homology 2 (MH2) domains to bind to DNA and interact with co-repressors and co-activators, respectively (Shi et al, 1998; Feng & Derynck, 2005). Known co-regulators that guide the SMAD complexes to distinct promoter sequences and regulate target gene expression include TFs such as ATF3, RUNX1, TGIF, SKIL, or SKI, and epigenetic regulators such as HDAC1/3/4/5/6, and GCN5 (Miyazawa et al, 2002; Ross & Hill, 2008). Upon TGFβ stimulation, SMADs can induce the expression of their collaborating TFs. JUNB, RUNX1, SNAI1, and SNAI2, thereby establishing feedforward loops (FFL), in which SMAD controls target genes both directly via promoter binding and indirectly via co-factor up-regulation (Peinado et al, 2003; Sundqvist et al, 2013, 2018).

Current evidence suggests that the temporal dynamics of SMAD signaling play an important role in cellular decision-making. Using time-lapse microscopy of individual MCF10A cells, we found that

---

[1]Department of Systems Biology, Institute for Biomedical Genetics (IBMG), University of Stuttgart, Stuttgart, Germany   [2]Stuttgart Research Center for Systems Biology (SRCSB), University of Stuttgart, Stuttgart, Germany   [3]Department of Biology, Technische Universität Darmstadt, Darmstadt, Germany   [4]Department of RNA-Biology & Bioinformatics, Institute for Biomedical Genetics (IBMG), University of Stuttgart, Stuttgart, Germany

Correspondence: loewer@bio.tu-darmstadt.de; legewie@ibmg.uni-stuttgart.de
Kolja Becker's present address is Boehringer Ingelheim, Biberach, Germany
Stefan Bohn's present address is AbbVie, Ludwigshafen am Rhein, Germany

---

transient SMAD2 dynamics are associated with increased cell motility, whereas sustained SMAD2 dynamics, in addition, induce cell cycle arrest and further reinforced cellular motility toward EMT (Strasen et al, 2018; Bohn et al, 2023). Comparable results were observed in pancreatic cancer cell lines, where a sustained SMAD signal was required for cell cycle arrest (Nicolás & Hill, 2003). In addition, in myoblasts TGFβ pulses as opposed to continuous stimulation, induced target gene expression and fate regulation (Sorre et al, 2014). Studies on other signaling pathways have shown that the strength and duration of intracellular signals generally play a pivotal role in influencing the decisions cells make regarding their fate, for example, in MAPK-induced proliferation versus neuronal differentiation of PC12 cells, or during p53-induced DNA repair versus cellular senescence (Marshall, 1995; Behar & Hoffmann, 2010; Purvis et al, 2012; Purvis & Lahav, 2013). To induce specific cell fates, different gene expression programs need to be induced depending on the temporal dynamics of signaling. Prior studies combined RNA sequencing with mathematical modeling to investigate EMT regulators upon TGFβ-signaling (Deshmukh et al, 2021). However, it has been a matter of active research on how gene expression networks distinguish input signals, for example, through epigenetic regulation mechanisms and feedback and feedforward regulation by TFs resulting in different phenotypic outcomes (Mangan & Alon, 2003; Weidemüller et al, 2021). For instance, systems biological approaches combining quantitative experiments with dynamic modeling have shown that coherent FFL, where both inputs directly and indirectly induce gene expression networks, allow for decoding the amplitude and duration of input signals (Alon, 2007). Furthermore, signal decoding at the level of target gene expression may be related to mRNA half-life (Uhlitz et al, 2017).

For TGFβ-SMAD signaling, systems biological studies investigated the molecular mechanisms shaping temporal dynamics of signal transmission from the cell membrane to the nucleus, or focused on the downstream target gene expression level (Zi et al, 2011; Zi et al, 2012; Strasen et al, 2018). For instance, by combining live-cell imaging and stochastic modeling, Molina et al investigated the impact of various stimuli including TGFβ on gene expression kinetics at the CTGF locus (Molina et al, 2013). Similarly, Frick et al and Tidin et al studied the importance of relative changes (fold changes) in the SMAD signal for the gene expression of SNAI1 and CTGF target genes at the single cell level (Frick et al, 2017; Tidin et al, 2019). Extending on a larger set of ~12 prototypical target genes, Lucarelli et al demonstrated that differential SMAD complex formation is predictive for the regulation of downstream target gene expression in hepatocytes (Lucarelli et al, 2018). Despite these insights into SMAD signaling and gene expression, we are currently lacking quantitative, systems-level insights into the relationship between SMAD signaling dynamics, global gene expression, and cell fate decisions.

In this work, we obtained a comprehensive view of how the TGFβ-induced gene-regulatory network governs cell fate decisions in mammary epithelial MCF10A cells, a well-established cellular model for EMT regulation by TGFβ. We experimentally characterized the response at multiple levels, including live-cell imaging of SMAD signaling, time-resolved genome-wide RNA sequencing and EMT assays. Using mathematical modeling, we classified target genes into potential direct targets of SMAD TFs, for which SMAD may serve as a sole input, and into indirect targets, for which regulation comprised

the activity of other TFs in complex feedback and feedforward interaction. Interestingly, target genes belonging to the latter group frequently show a strong delay in gene expression, discriminate between TGFβ doses and are highly relevant for cell fate decisions such as EMT. Using TF knockdowns (KDs), we identify regulation mechanisms controlling delayed target genes: SNAI1, SNAI2, SKIL, SKI, and RUNX1 function as early repressors of SMAD-dependent gene regulation, preventing premature activation, whereas JUNB serves as a delayed activator, boosting expression at late time points. We confirm that JUNB is an important enhancer of delayed gene expression upon strong stimulation, thereby specifically inducing EMT at high, but not low, TGFβ doses. Taken together, we use a genome-wide systems biology approach to determine key regulatory candidates involved in SMAD signal decoding, and cell fate decisions.

# Results

## Signaling and gene expression is dose-dependent in response to TGFβ

To characterize TGFβ-signaling and downstream gene expression, we used a previously established MCF10A reporter cell line with histone H2B cyan fluorescent protein tag as a nuclear marker (H2B-CFP), and SMAD2 yellow fluorescent protein tag (SMAD2-YFP). As observed in earlier live-cell time-lapse imaging experiments, SMAD2 resides in the cytosol in the absence of stimulation and shows nuclear translocation upon stimulation with low (2.5 pM) and high (100 pM) doses of TGFβ (Strasen et al, 2018). Treatment with 2.5 pM results in a lower early signal amplitude 60 min after stimulation when compared with 100 pM. Moreover, the signal is sustained for the high dose, whereas it is transient for the low dose, reaching baseline ~720 min post-stimulation (Fig 1A). In our earlier published work, we have shown that these TGFβ-induced nuclear translocation dynamics of the reporter system reflect the behavior of endogenous SMAD2 protein using immunofluorescence (Strasen et al, 2018). Furthermore, we observed that the signaling activity determined by the reporter system closely mirrors SMAD phosphorylation dynamics (Fig S1A) (Strasen et al, 2018; Bohn et al, 2023).

Under the same stimulation conditions, we performed bulk RNA sequencing at various time points (45/90/180/360/720/1,440 min) post-stimulation to assess global changes in gene expression. In total, 4,823 genes were differentially regulated in at least one TGFβ dose and time point (P-value < 0.01, multiple testing corrected) (Fig 1B). Approximately, one-third of the genes were up-regulated by TGFβ, whereas the remainder was down-regulated (Fig 1C). As expected, the vast majority (2,027 genes, 90%) of the 2,262 genes responding at the low dose were also differentially expressed upon strong stimulation. On top, the 100 pM stimulus specifically regulated a large set of 2,561 additional genes that did not show significant changes upon treatment with 2.5 pM (Fig 1B). Hence, the higher the TGFβ stimulus, the more genes are significantly regulated. In addition, the magnitude of mRNA up- or down-regulation increases in a dose-dependent gradual manner, as indicated for known TGFβ target genes (Fig 1D) and in a heatmap showing the trajectories of all 4,823 differentially expressed genes (Fig 1C).

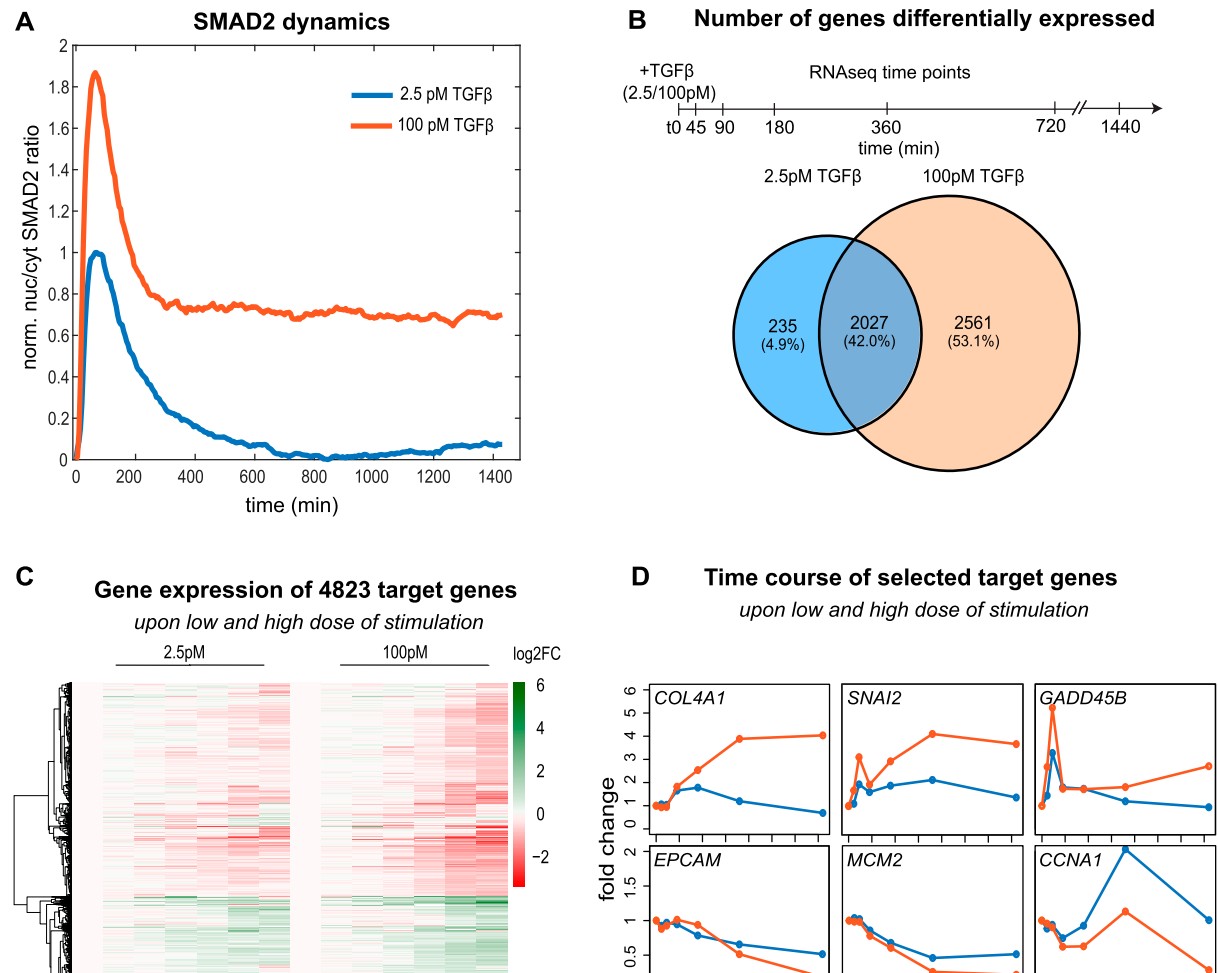

**Figure 1. Dose-dependent signaling and gene expression in response to TGFβ.**

**(A)** Time course of normalized nuc/cyt SMAD2 ratio upon stimulation with a low (2.5 pM, blue) and high (100 pM, orange) TGFβ dose. Shown is a background-subtracted mean of the cell population derived from the imaging data in Strasen et al (2018); Strasen et al (2018) of 378 cells (100 pM), and 351 cells (2.5 pM). **(B)** Venn diagram shows a number of genes differentially expressed upon 2.5 pM- and 100-pM TGFβ stimulation in MCF10A cells, in any of the time points of the sketched RNA-sequencing experiment. In total, 4,823 genes show altered expression compared with unstimulated control, of which 235 (~5%) and 2,561 (~53%) specifically respond at 2.5 and 100 pM, respectively, whereas 2,027 (~42%) change expression in both doses. **(C)** Heatmap shows trajectories of all 4,823 differentially expressed genes upon low and high dose, genes (y-axis) being sorted by hierarchical clustering. **(D)** Time course of known TGFβ target genes with biological functions related to EMT (COL4A1, SNAI2, EPCAM) and cell cycle (GADD45B, MCM2, CCNA1).

## Late target genes specifically respond at high TGFβ doses

Given this dose-dependence, the question arises whether the same set of genes is regulated by both TGFβ doses, or whether a subset of genes is specifically controlled upon sufficiently strong stimulation. To answer this question, we related log₂-fold changes of the low and high doses at each time point, considering all genes that are differentially regulated in at least one of the two conditions (Figs 2A and S2A). As reported in our earlier work (Bohn et al, 2023), we find little specificity of TGFβ-induced gene expression at early time points (90/180/360 min), as essentially the same genes are induced by the low and the high dose, with a

global trend showing larger up-regulation at 100pM TGFβ (Fig 2A, solid line).

Similarly, at the late 720 and 1,440 min time points, most genes still show coherent regulation by the two doses, closely following the global trend line. However, a subset of TGFβ-induced mRNAs shows atypical behavior, being strongly (up to 64-fold) up-regulated upon 100pM TGFβ stimulation, when showing essentially no response to the 2.5 pM stimulus. To comprehensively characterize these late dose-discriminating genes (DDGs), we applied filtering to the dataset, requiring that a gene shows a small fold change (<twofold) at the low dose when showing strong regulation (>fourfold) at the high dose 1,440 min after stimulation.

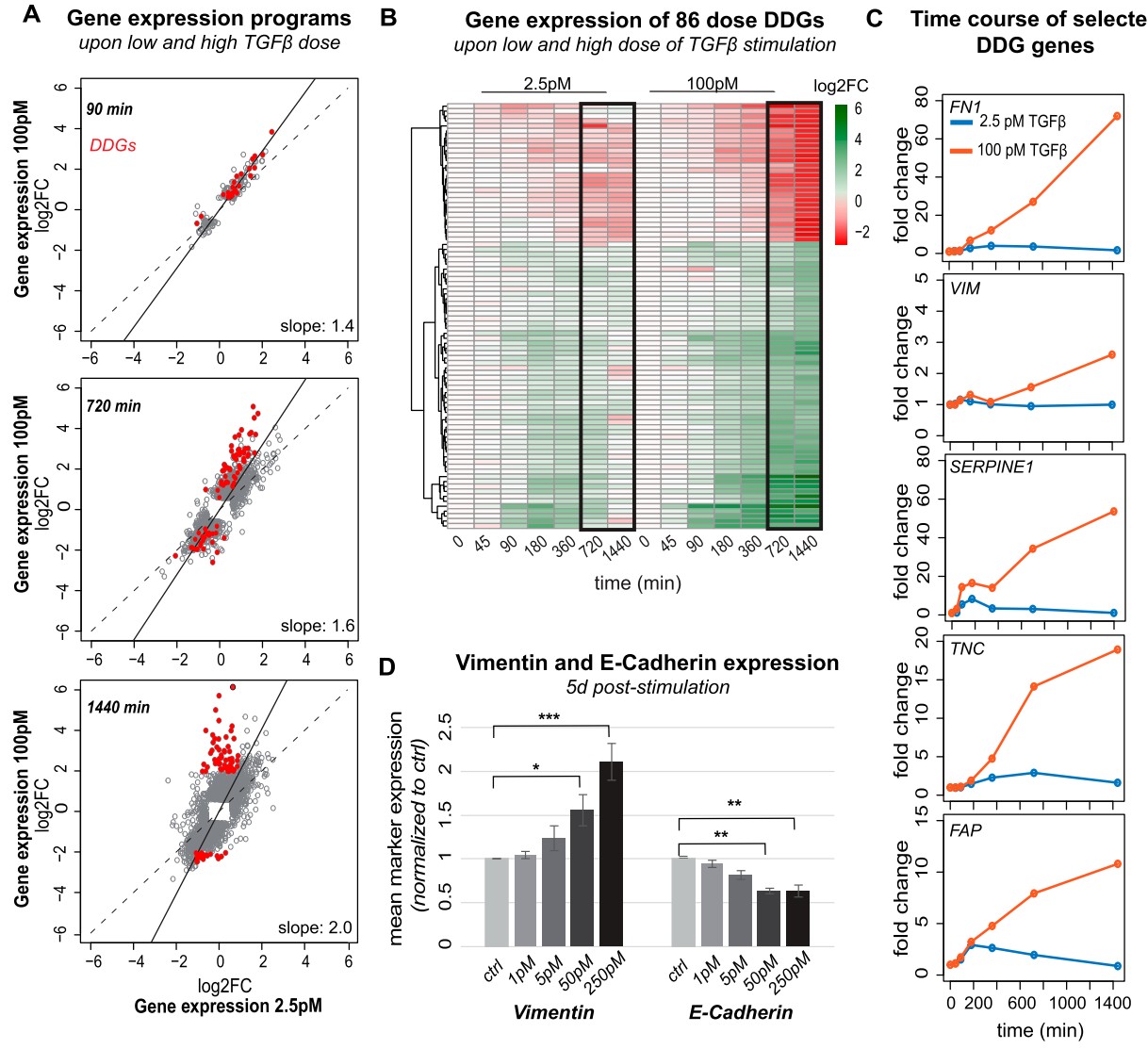

**Figure 2. Dose-discriminating genes (DDGs) specifically respond at high TGFβ doses at late time points and are related to EMT.**
**(A)** Scatter plot relating global gene expression changes upon 2.5- and 100-pM TGFβ stimulation at different time points (90/720/1,440 min) in MCF10A cells. Each grey dot represents a significant differentially expressed gene in at least one of the two stimulation conditions, and the black solid line is a linear fit to the data. Red dots indicate DDGs differentially expressed upon high-dose stimulation (absolute [abs.] fold change [FC] > 4) but not low-dose stimulation (abs. FC < 2) at late time points (720 or 1,440 min post-stimulation). Number of genes: 90 min: DDGs: 24, all: 187, 720 min: DDGs: 84, all: 3,009, 1,440 min: DDGs: 86, all: 3,920. **(A, B)** Heatmap shows trajectories of the set of 86 DDGs defined in (A). **(C)** Time course of selected DDGs (FN1, VIM, SERPINE1, TNC, FAP) upon low- and high-dose TGFβ stimulation. **(D)** EMT is specifically induced at high TGFβ doses. Protein levels of vimentin and E-cadherin measured by flow cytometry upon 5 d stimulation with different TGFβ doses (1/5/50/250 pM) normalized to unstimulated control. Bars comprise different number of biological replicates (n = 3 for 5 pM, n = 6 for ctrl, 1/50/250 pM), errors indicate SEM and significance level relative to unstimulated control (*$P \leq 0.05$, **$P \leq 0.01$, ***$P \leq 0.001$).

This yielded a set of 86 DDGs, which were subjected to further analysis (Fig 2B and C).

Interestingly, the DDGs were associated with the regulation of EMT and therefore significantly overlapped with a published EMT gene set (Fisher's exact test, $P$-value$<2 \times 10^{-16}$) (Table S1). Specific examples of DDGs are well-known TGFβ-induced EMT genes, such as FN1, SERPINE1, and VIM, but also the cell cycle regulator TNC (Fig 2C). Gene-set-enrichment-analysis of DDGs indicated an overrepresentation of genes involved in extracellular-matrix remodeling, focal adhesion formation, and regulation of proteoglycans, processes associated with increased cell movement during EMT (Table

S2). To validate that EMT is specifically or at least more strongly induced by the high TGFβ dose, we conducted a well-established flow cytometry–based EMT assay, measuring the up- and down-regulation of vimentin and E-cadherin protein levels, respectively. In line with our hypothesis, the flow-cytometry data shows that only higher doses of TGFβ (50–250 pM) significantly induce EMT of MCF10A cells, whereas lower doses of 1–5-pM TGFβ cannot initiate the mesenchymal phenotype 5 d post-TGFβ stimulation (vimentin high, E-cadherin low) (Fig 2D). In our previous work, we had shown that the conversion to the vimentin high, E-cadherin low state is accompanied by increased migration and therefore reflects EMT in

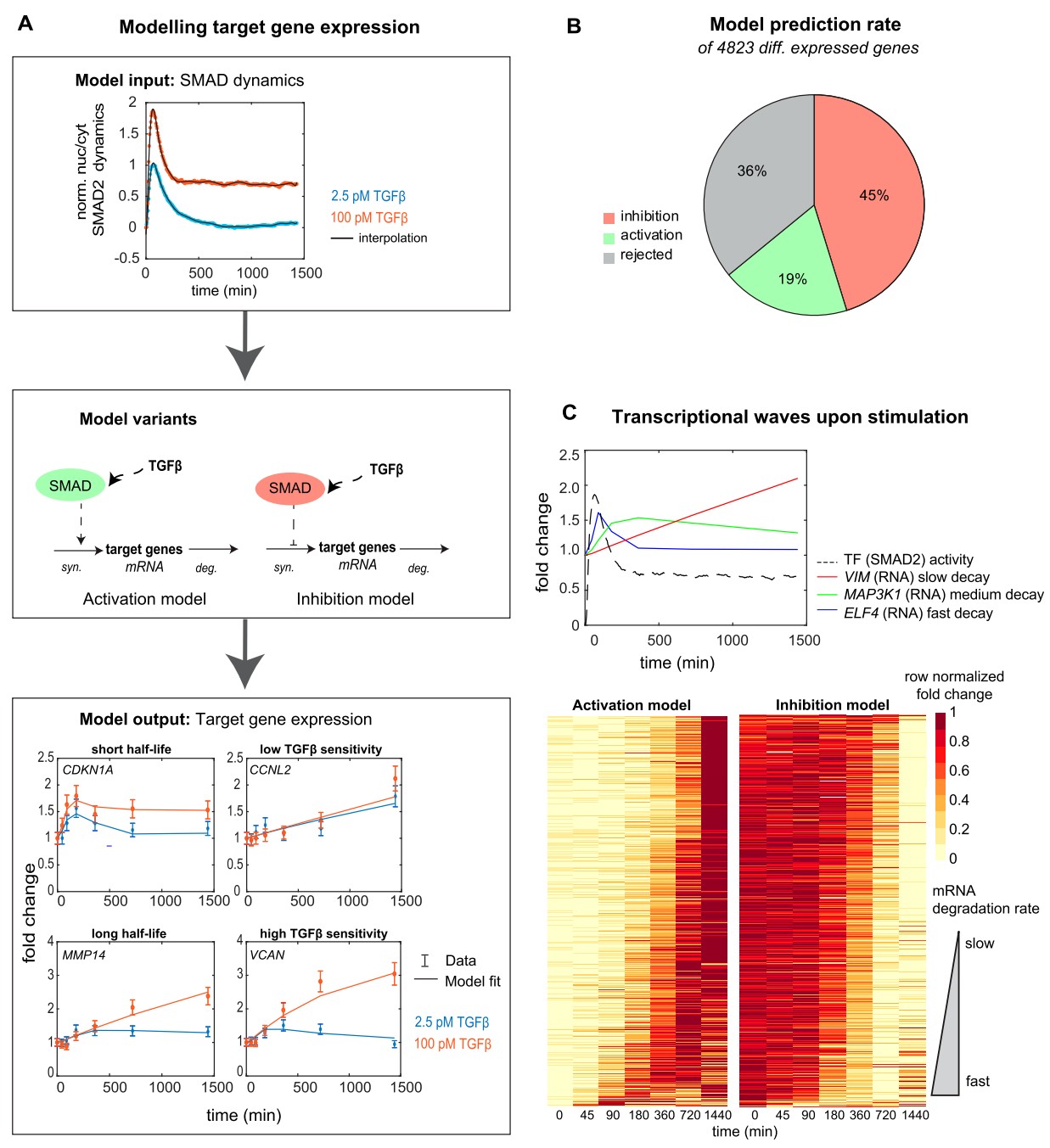

**Figure 3. Modeling SMAD-dependent activation and repression of gene expression.**
**(A)** Modeling framework: experimentally measured SMAD2 dynamics (top) serve as model input, controlling target gene transcription in an ODE describing mRNA synthesis and degradation (middle). In two model variants, target gene activation and repression by SMAD are considered, and the simulated RNA time course is fitted to the RNA-sequencing data (bottom). The ODE model describes target genes with short (CDKN1A) and long half-lives (MMP14), and by adjusting the Hill equation parameters genes with low (CCNL2) and high (VCAN) TGFβ sensitivity. **(B)** Proportions of explained and rejected target genes. 64% (3,091 genes) of 4,823 target genes can be explained by either activation (19%, 919 genes) or inhibition (45%, 2,160 genes) model, whereas 36% (1,744 genes) are rejected as judged by $\chi^2$-testing of model fitting results (see the Materials and Methods section). **(C)** Model predicts the order of target gene expression based on mRNA degradation rates. Top: simulated time courses using the best-fit parameters of three exemplary genes, one with fast degradation (ELF4, blue), one with medium degradation (MAP3K1, green), and one with slow degradation (VIM, red). Bottom: heatmap showing measured time courses of target genes explained by the activation (919 genes) and inhibition (2,160 genes) models after sorting according to the best-fit mRNA degradation rate. Each row is min–max normalized by its minimal and maximal expression values.

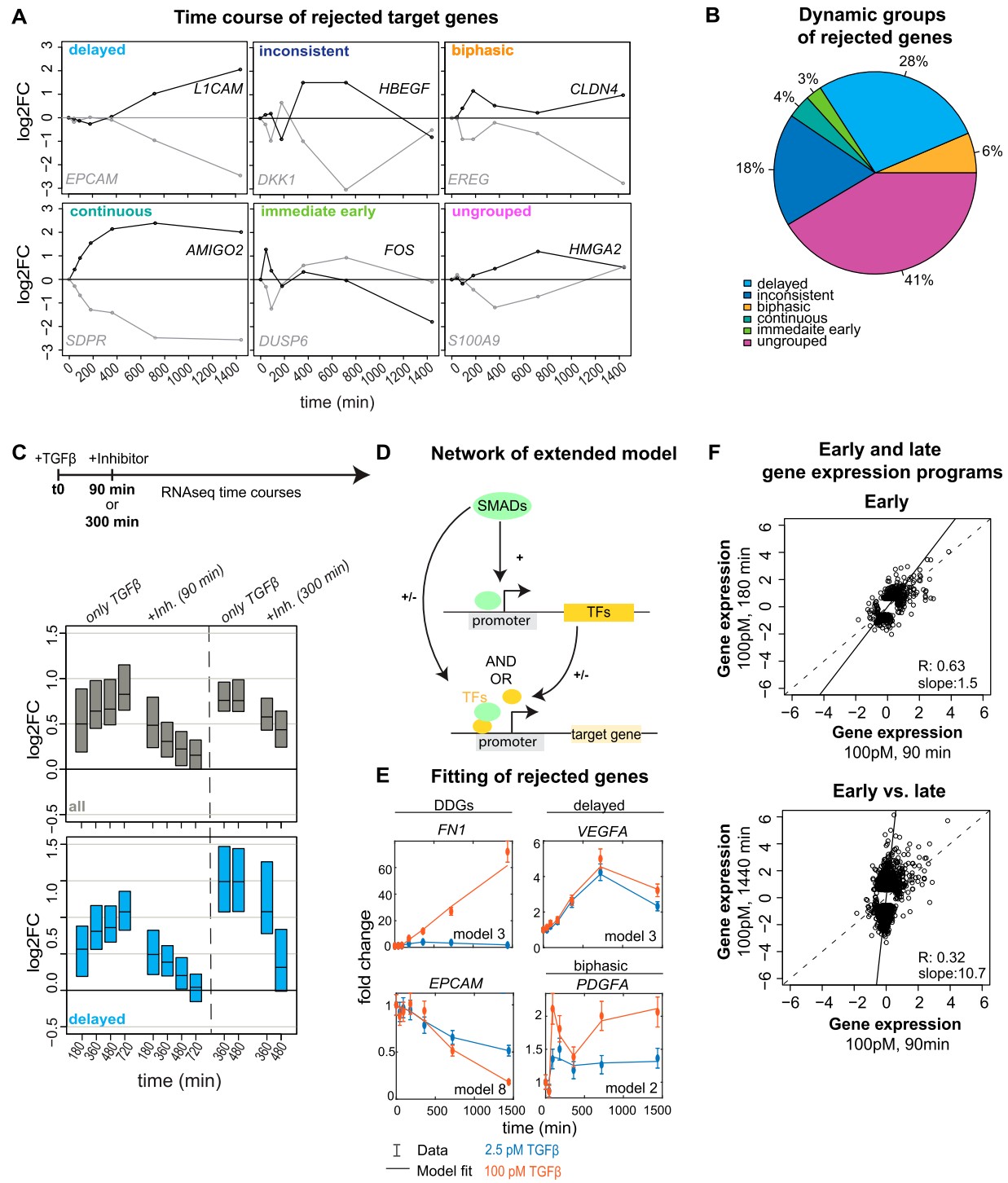

**Figure 4. Evidence for feedforward regulation in SMAD-dependent gene expression.**
**(A)** RNA-sequencing time courses (100pM TGFβ) of selected genes rejected by the kinetic models in Fig 3, gene names being indicated next to each trajectory. The following temporal dynamics can be observed: delayed (late onset of differential expression), inconsistent kinetics (early up-regulation and late down-regulation, or vice versa), biphasic (fast and sustained differential expression with intermediate decline), continuous (fast and sustained differential expression), and immediate early (early differential expression followed by rapid decline). Ungrouped genes, such as S100A9 and HMGA2 show distinct dynamics and/or low expression changes and can therefore not be assigned to any of the dynamic groups. **(B)** Automated classification of all 1,744 rejected genes into the five-time course patterns (immediate early, biphasic, continuous, delayed, and inconsistent kinetics) using expression cutoffs at multiple time points (see the Materials and Methods section and Table S3). **(C)** RNA sequencing after TGFβ-receptor–inhibitor treatment confirms ongoing SMAD-dependency of late target genes. Cells were stimulated with 100pM TGFβ at t = 0 and the TGFβ-receptor inhibitor SB431542 was applied at 90 or 300 min. Genome-wide RNA sequencing was performed at the indicated time points, either in cells treated with TGFβ only (left) or after inhibitor addition (right). The expression distributions (boxes: median with lower and upper quartile) of all up-regulated (log₂FC > 0.58, adj. *P*-value < 0.01) TGFβ target genes (top, grey, + 90 min: 789 genes, + 300 min: 993 genes) or those belonging to the delayed gene expression group (bottom, blue, + 90 min: 112

MCF10A cells (Bohn et al, 2023). To further validate that low TGFβ doses fail to induce EMT, we performed an extensive 11 d time course experiment at multiple doses and found that the spindle-shaped cell morphology, which is characteristic for mesenchymal cells, is observed for 100pM TGFβ stimulation, but absent at low TGFβ doses (Fig S1B). Therefore, only a sustained SMAD2 signal, observable at high TGFβ doses using live-cell imaging (Fig 1A) and by phospho-SMAD2 immunoblotting (Fig S1A), induces EMT, whereas a transient signal fails to do so.

Inspection of the time courses of dose-discriminating EMT and cell cycle genes (FN1, SERPINE1, VIM, TNC) revealed exceptionally high fold changes 720 and 1,440 min after 100pM TGFβ stimulation, with much lower or no mRNA induction in the first few hours. Such a pronounced induction delay is particularly observed for FN1, VIM, TNC, FAP, and to some extent, also for SERPINE1, which shows a transient up-regulation at early time points, and a more pronounced second rise 720 and 1,440 min after stimulation (Fig 2C). Thus, DDGs show a late expression boost with high fold changes, whereas they return to basal within 1,440 min upon stimulation with the low dose. Taken together, a subset of EMT and cell cycle relevant targets shows a highly distinct expression profile for varying TGFβ doses late after stimulation, which may boost these cell fates, especially upon strong stimulation.

### Dynamics of most genes are described by a simple gene expression model

Because the temporal dynamics of gene expression are shaped by many aspects including mRNA half-life, promoter saturation, and the interplay of SMADs with other TFs, we turned to mathematical modeling for a quantitative description of mRNA time courses on a genome-wide scale. In our basic gene expression model, we describe the mRNA time course using a single ordinary differential equation (ODE), which takes into account rates of mRNA synthesis and degradation (Fig 3A). The mRNA synthesis rate is assumed to depend on the nuclear SMAD2 activity (SMAD) in a sigmoidal manner (Hill equation), and the model, in addition, considers basal, SMAD-independent transcription ($\beta_0$). As SMAD TFs may induce or repress target gene expression, we considered two variants of this model, an activator Equation (1) and an inhibitor Equation (2) model

$$\frac{dy}{dt} = \beta_0 + k_{syn} * \frac{SMAD^h}{SMAD^h + k^h} - \beta_0 * y \qquad (1)$$

$$\frac{dy}{dt} = \beta_0 * \left(1 - \frac{SMAD^h}{SMAD^h + k^h}\right) - \beta_0 * y \qquad (2)$$

where $y$ is the fold change in target gene expression, $\beta_0$ the basal transcription and degradation rate, $k_{syn}$ the maximal transcription rate, SMAD the SMAD2 input signal, $h$ the Hill coefficient, and $k$ the concentration of promoter half-saturation.

To assess whether these simple models agree with our data, we used the measured SMAD2 nuclear translocation as a model input and fitted the simulated mRNA fold change to the corresponding RNA-sequencing measurements of each gene, adjusting Hill equation parameters and the mRNA degradation rate ($\beta$0). For each of the 4,823 differentially expressed genes, the activator and inhibitor models were tested separately, simultaneously considering the nuclear-to-cytoplasmic (nuc/cyt) SMAD2 ratio and RNA-sequencing data at low and high TGFβ doses (Fig 3A). Based on the fitting results, we concluded that 64% of all differentially expressed target genes could be quantitatively explained by one of the two models, 19% and 45% being described by the activator and inhibitor version, respectively (chi-square [$\chi^2$] test, see the Materials and Methods section) (Fig 3B). Taken together, minimal SMAD2-dependent mRNA expression models account for a large part of the global gene expression program, suggesting that much of dose-dependence simply follows gradual changes in SMAD2 activity, and does not require additional input by other signaling pathways or TFs.

The dynamics of the explained target genes show a large variation, some mRNAs react fast, whereas others do so only with a delay (Fig 3A and C). Our model accommodated this different kinetics by adjusting mRNA half-life: genes with a short, fitted mRNA half-life rapidly follow the transient SMAD2 increase and decline (e.g., CDKN1A), whereas long-lived mRNAs show a late response and remain differentially expressed long after the SMAD2 input has vanished (e.g., MMP14) (Fig 3A). Hence, in line with previous reports (Hao & Baltimore, 2009), the mRNA degradation rate is the major controller of the temporal order of gene expression (Fig 3C). In contrast, the Hill equation parameters primarily determine TGFβ dose sensitivity, as some mRNA show a strong difference between the low and the high TGFβ doses (e.g., CDKN1A, MMP14, VCAN), whereas others show saturation already at the low TGFβ dose (e.g., CCNL2) (Fig 3A). Taken together, our modeling framework classifies genes according to mechanisms of regulation and provides mechanistic explanations for the dynamics of the explained genes.

### Genes with complex regulation show signs of feedforward regulation

Of 4,823 differentially expressed genes in response to TGFβ stimulation, 1,744 (36%) could not be explained by the simple SMAD-dependent ODE models, and therefore show signs of complex

genes, + 300 min: 88 genes) declines over time, upon inhibitor treatment. See SF3 for analysis of down-regulated genes and other gene groups. **(D)** Schematic representation of extended feedforward loop (FFL) model. In the FFL model, SMAD up-regulates (+) the expression of a hypothetical TF (TF), which in turn regulates target genes jointly with SMAD. Both SMAD and the TF function either as a repressor (−) or activator (+), and control on the target gene promoter with AND- or OR-gate logic, giving rise to in total eight model variants (see the Materials and Methods section for details and Fig S4G for model variants and Table S5 for FFL fitting results). **(E)** Time course fits of selected DDGs (FN1, EPCAM), delayed (VEGFA), and biphasic (PDGFA) target genes with optimal extended model version indicated on the bottom right. **(F)** Distinct early and late gene expression programs induced by TGFβ. Scatter plot relating log₂-fold changes relative to unstimulated control of two early time points (top, 90 min versus 180 min) or early versus late time points (bottom, 90 min versus 1,440 min). Each dot represents a significant differentially expressed gene in at least one stimulation condition (top: 516 genes, bottom: 3,722 genes). Lines: bisecting (dashed) and linear fit to the data (solid), slope and correlation coefficient indicated on the bottom right.

regulation. Many of these 1,744 rejected genes seem to be important for cellular decision-making, as they contain 156 of 391 differentially regulated cell cycle genes, and 93 of the 172 differentially regulated EMT genes (Table S3). Manual inspection of time courses identified five recurrent time course patterns (Fig 4A and B):

 (I) Delayed kinetics: strong rise, but with a pronounced delay
 (II) Inconsistent kinetics: early up-regulation followed by late down-regulation (or vice versa)
(III) Biphasic kinetics: transient expression followed by a second, late rise (or drop)
(IV) Continuous kinetics: continuous up- or down-regulation despite a transient SMAD input
 (V) Immediate early kinetics: early up-regulation followed by rapid decline ("spike like")

By time course filtering (see the Materials and Methods section and Table S4), we could assign a large number of 1,021 of 1,744 rejected genes to one of these five dynamic gene groups, the majority belonging to the delayed (483), inconsistent (315), and biphasic (112) classes (Figs 4A and B and S3A–E, Table S4). The remaining 723 of 1,744 rejected genes, not assigned to any of the dynamic gene groups (i–v), tend to show minor induction or repression by TGFβ (<2.5-fold), especially at late time points (Fig S3F). Taken together, many genes rejected by our simple kinetic models show signs of late expression regulation after the SMAD signal has reached plateau (delayed, biphasic, inconsistent, and continuous groups). In line with late gene expression regulation being important for TGFβ dose discrimination (Fig 2), the 55 DDGs that could not be described by the SMAD-dependent gene expression models were classified into delayed (33 of 55), continuous (21 of 55), and biphasic (1 of 55) classes (Fig S2B and C).

Previous studies suggested that late target gene regulation by TGFβ involves FFLs, in which the input (SMAD2) regulates late target genes by a combination of direct and indirect mechanisms, that is, direct SMAD binding to the gene promoter and by controlling the expression of transcriptional co-regulators (Sundqvist et al, 2018). In line with such feedforward regulation, many early target genes expressed within the first 90 min upon 100pM TGFβ stimulation encode for TFs and epigenetic regulators (39 of 109 differentially expressed genes, 39%) (Fig S4A and B). To test whether late target genes still directly depend on the initial SMAD signal, we treated cells 90 or 300 min after 100pM TGFβ stimulation with SB431542, a small molecule inhibitor of TGFβ receptors, which quickly down-regulates SMAD signaling with a half-life of ~60 min (Inman et al, 2002; Strasen et al, 2018). Following gene expression over time using RNA sequencing, we observed a quick down-regulation of almost all TGFβ target genes within 60–120 min of SB431542 application when compared with cells that had not been treated with the inhibitor. This suggests a continuing dependence on the SMAD signal, which importantly was also observed for late target gene sets belonging to the classes with delayed, continuous, or biphasic kinetics, and for those showing dose discrimination (Figs 4C and S4D–F).

To further validate large-scale feedforward regulation of the 1,744 rejected target genes, we extended our kinetic models of gene expression by a hypothetical SMAD-regulated TF (Fig 4D). This TF is described by the synthesis-degradation model (see the Materials and Methods section) and, in turn, regulates the synthesis of the target gene of interest. For model fitting, we again used the SMAD input and target gene measurements at low and high TGFβ doses, when leaving the dynamics of the unknown TF unconstrained. Because both SMAD2 and TF may function as activators or repressors of gene expression, and they may interact on target gene promoters with AND- or OR-logic, we considered in total eight model variants (Fig S4G). After fitting and statistical testing, we found that 1,577 of 1,744 initially rejected target genes could be quantitatively explained by the extended FFL models (Fig 4E, Table S5). Of these, 1,074 of 1,577 (68%) were explained with the co-TF acting as an activator, whereas for 503 of 1,577 (32%) target genes the co-TF functioned as a repressor. When combined, the minimal model and its FFL derivatives explained the expression of 4,656 of 4,823 (96.5%) SMAD target genes, suggesting that the late TGFβ-induced gene expression program is shaped by both repressors and activators of target genes.

As feedforward regulation may lead to the up-regulation of distinct genes late after stimulation, we tested whether gene expression is reprogrammed over time. To this end, we globally related fold changes in target gene expression across time points for the 100pM TGFβ stimulus, and indeed found highly distinct gene expression programs between early (90 min) and a late (720/1,440 min) time points, many genes showing specific expression late after stimulation. In contrast, the response at two early time points (90 and 180 min) was highly similar (Figs 4F and S4C). Moreover, even though TFs and epigenetic regulators were also induced early upon 2.5-pM TGFβ stimulation (22 of 53 differentially regulated genes, 41%) (Fig 4A and B) their average fold change was significantly lower compared with 100-pM TGFβ. Consistently, the early and late gene expression programs at 90 and 720 min stimulation were highly similar at the low dose (Fig S4C), confirming specific late transcriptional reprogramming at high TGFβ doses.

## Feedforward regulation by early repressors and JUNB as a late activator determines DDG expression

To identify potential regulators of late gene expression, we turned to TF KDs followed by global RNA sequencing. For the identification of candidates, we used a curated list of TFs and epigenetic regulators (Lambert et al, 2018), and focused on those induced within the first 90 min after TGFβ stimulation, thereby obtaining a shortlist of seven co-factors (SNAI2, JUN, JUNB, KLF10, SKIL, RUNX1, SKI) (Fig 5A). In addition, we included SMAD2 and SMAD3 KDs as positive controls and, furthermore, considered SNAI1 and ATF3, which are known induced TFs upon TGFβ stimulation but did not reach significance in our data set (Kang et al, 2003; Yin et al, 2010; Zhang et al, 2014).

For KD experiments, cells were pre-incubated with siRNAs for 24 h and then subjected to stimulation with a saturating concentration of TGFβ (100 pM) (Fig 5B). Samples were taken at the time point of stimulation (t0) or 90, 180, 360, and 720 min after stimulation. For each of the KDs, we validated highly efficient depletion of the targeted TF (Fig S5A) and performed bulk RNA sequencing to determine global changes in gene expression. The KD of SMAD2, SNAI1, SNAI2, and SKI had pronounced effects on basal gene

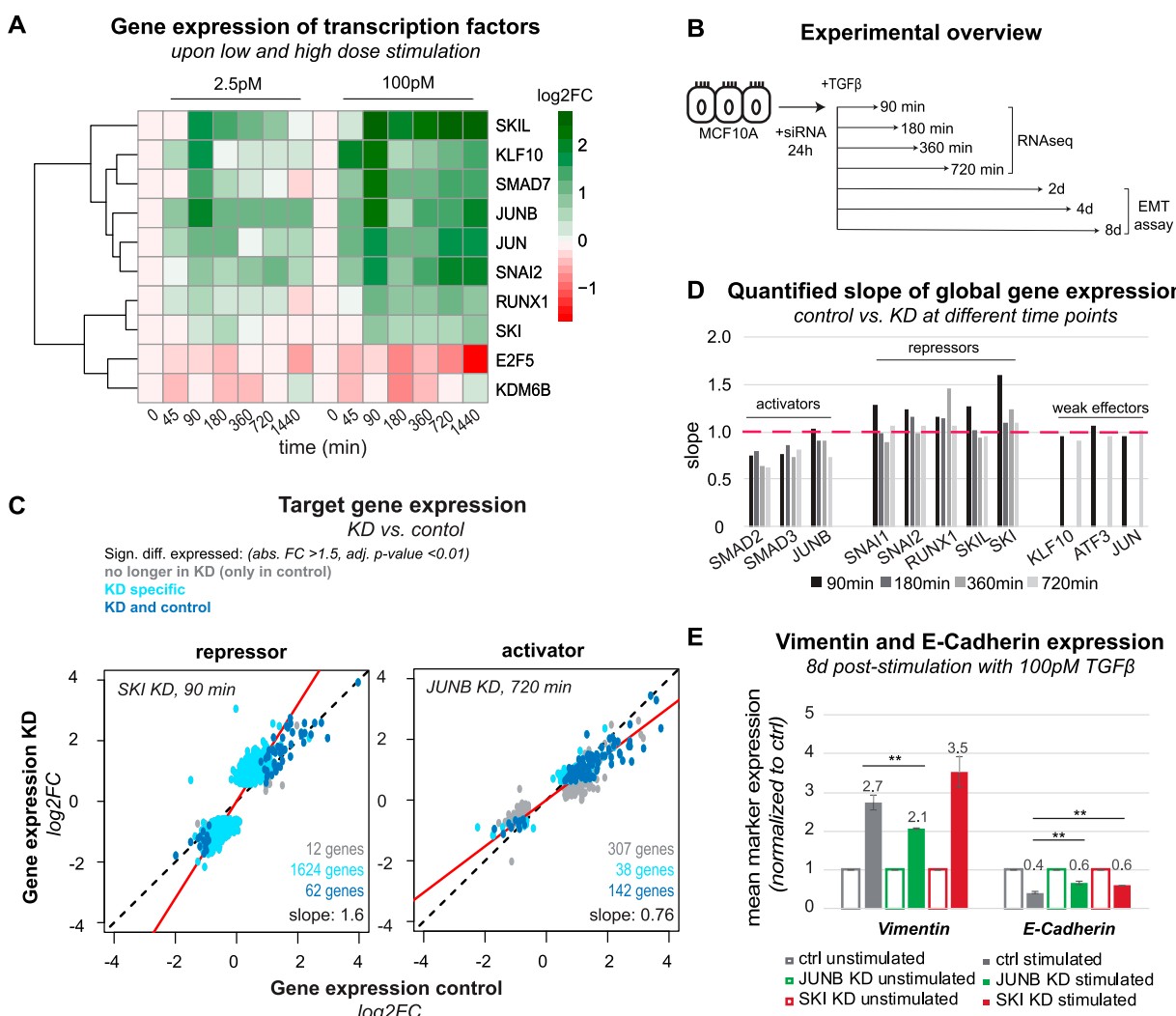

**Figure 5. TFs shaping TGFβ target gene expression and cell fates.**
**(A)** Gene expression time course of TFs (SKIL, KLF10, SMAD7, JUNB, JUN, SNAI2, RUNX1, SKI, E2F5), and epigenetic regulators (KDM5B) with differential expression (adj. *P*-value < 0.01, abs. FC > 1.5) in the first 90 min after TGFβ stimulation. **(B)** Experimental setup showing time points of RNA sequencing (90, 180, 360, and 720 min) and EMT assays (2, 4, and 8 d) post-stimulation with 100pM TGFβ of MCF10A cells pre-incubated with siRNA targeting TFs for 24 h. **(C, D)** KD reveals activators and repressors of SMAD-mediated gene regulation. **(C)** Scatter plot showing TGFβ-induced global gene expression changes in control versus SKI (90 min, left) and control versus JUNB (720 min, right) KD. The slope of a linear trend line (red line) across all genes determines the strength of the repressor (left) or activator (right) effect. Genes significantly differentially expressed (adj. *P*-value < 0.01, abs. FC > 1.5) in control only, KD only or control and KD are colored in grey, blue, and turquoise, respectively. **(C, D)** Quantified slope of KD effect on global gene expression, as described in (C), for the complete set of tested TFs and all time points (90, 180, 360, 720 min). SMAD2 and SMAD3 are activators at all time points, whereas JUNB is a late activator; repressors SNAI1, SNAI2, RUNX1, SKIL, SKI mainly act at early time points. KLF10, ATF3, and JUN show weak effects. **(E)** Flow-cytometry analysis of epithelial (E-cadherin) and mesenchymal (vimentin) marker expression upon long-term (8 d) TGFβ stimulation of control and KD (SKI and JUNB KD) MCF10A cells, n = 3, SEM is shown (**\*\**P ≤ 0.01). JUNB KD represses and SKI KD enhances the mesenchymal phenotype.

expression before stimulation, with 485, 846, 1,254, and 268 genes showing an absolute fold change greater than 1.5, respectively, whereas basal effects were negligible for the remaining factors (JUNB, SKIL, RUNX1, SMAD3, JUN, KLF10, and ATF3) (Fig S5B).

To quantify KD effects on TGFβ-induced gene expression, we related the log$_2$-fold changes induced by TGFβ stimulation in control versus KD cells across all genes (Fig 5C and D). In general, we found a strong correlation of TGFβ-induced log$_2$-fold changes (R = 0.87–0.99), suggesting that, in most cases, a similar overall gene expression program was activated by TGFβ in control and KD cells (Table S6). By fitting a linear trend line to the data, we determined

the direction of change and the magnitude of the KD effect on global target gene expression (Fig 5C, solid red line). TFs with slopes > 1 upon KD were considered as repressors, as their depletion leads to target gene up-regulation, whereas the opposite (slope < 1) is true for activators. Based on this criterion and statistical analysis, KDs of SNAI1, SNAI2, JUNB, SKI, SKIL, and RUNX1 as well as SMAD2, and SMAD3 had significant effects on global TGFβ-induced gene expression (*t* test, *α* = 0.05). We found that RUNX1, SKIL, SKI, SNAI1, and SNAI2 function as repressors (slope >1), whereas JUNB, SMAD2, and SMAD3 function as activators (slope <1) (Fig 5C and D; Table S6).

Given the KD analysis at different time points, we could determine the timing of TF action on target gene expression: As a general trend, the repressors primarily acted shortly after TGF$\beta$ stimulation, with SKI showing the strongest effect on target gene expression at 90 min among all repressors. Within the activators of TGF$\beta$ target gene expression, SMAD2 and SMAD3 had pronounced effects at all time points (90/180/360/720 min) as expected, whereas JUNB specifically acted at the late time point, consistent with the reported role of JUNB being a feedforward regulator of SMAD-induced gene expression that redirects SMADs to novel target genes (Sundqvist et al, 2018). To exclude that changes in gene expression upon TF KD arise from feedback to the SMAD signaling pathway, we examined the nuc/cyt translocation of SMAD2 upon KD of JUNB, SKIL, and SKI (Fig S5C and D, Table S7). We observed only slight alterations in SMAD2 translocation, excluding that feedback regulation is the major cause of changes in gene expression.

Some TFs tended to globally affect all target genes, whereas others have the most pronounced effects on a subset of target genes, therefore controlling a specific gene expression program (Figs 5C and S6). To quantify such TF specificity, we determined the number of genes that show a significantly different TGF$\beta$ response in the KD when compared with non-targeting control. For some KDs the TGF$\beta$ response of almost all target genes is homogeneously affected (SMAD2, SMAD3) (Fig S6), whereas for other KDs, only a subset of genes shows a differential TGF$\beta$ response and is observed for early repressors (SKIL, SKI, RUNX1, SNAI1, and SNAI2, 90–360 min) and JUNB as a late activator (720 min) (Figs 5C and S6, compare dark blue, light blue, and grey dots).

Taken together, the TFs decompose into repressors of SMAD-induced gene expression primarily acting at early time points (SKIL, SKI, RUNX1, SNAI1, SNAI2) and into activators that are either general mediators of the signal at all time points (SMAD2, SMAD3) or specifically act late (JUNB), possibly as feedforward amplifiers. Six factors seem to control a specific subset of SMAD target genes (SKI, SKIL, SNAI1, SNAI2, RUNX1, JUNB), indicating that they may reshape the SMAD-induced gene expression program. The remaining factors JUN, KLF10, and ATF3 had a weak impact on overall gene expression patterns, both before and after TGF$\beta$ stimulation (Fig S5B).

## SMAD co-factors modulate TGF$\beta$-induced EMT

To confirm that regulators of late gene expression affect cellular decision-making, we quantified EMT by measuring protein levels of vimentin and E-cadherin upon KD of a co-activator (JUNB) and a co-repressor (SKI) 2–8 d after stimulation with 100pM TGF$\beta$. To maintain the KD and to replenish TGF$\beta$ in these long-term experiments, we re-treated the culture with fresh siRNAs and TGF$\beta$ every 2 d, and confirmed that the down-regulation of the targeted TF was stable over time (Fig S7A).

Compared with unstimulated control, TGF$\beta$-induced EMT induction is weakened in JUNB KD cells, as evidenced by less pronounced vimentin up-regulation and E-cadherin down-regulation 2, 4 (Fig S7B and C) and especially 8 d post-stimulation (Fig 5E). This decreased sensitivity to EMT induction by TGF$\beta$ is consistent with JUNB being an activator of late TGF$\beta$-induced EMT genes, and agrees with an earlier report that showed reduced invasive behavior upon JUNB KD in three-dimensional spheroids, consisting of

MCF10A cells (Antón-García et al, 2023). On the contrary and as expected, the transcriptional repressor SKI blocks EMT induction, as vimentin was more up-regulated by TGF$\beta$ in SKI KD compared with control, 8 d after stimulation (Fig 5E). However, at earlier time points (2 and 4 d post-stimulation), SKI KD led to lowered vimentin induction by TGF$\beta$, suggesting that SKI might actually enhance the onset of EMT (Fig S7B). Taken together, our data place JUNB as an activator of EMT in MCF10A cells, whereas SKI may play a dual role in this process, exerting its repressive role only at late time points.

## JUNB reinforces the dose discrimination at the level of target gene expression

Given its role as an amplifier of late gene expression and EMT induction, JUNB may contribute to TGF$\beta$ dose discrimination, specifically boosting the late DDG expression upon high-dose stimulation. To confirm such late amplification, we compared time courses of selected DDGs (e.g., FN1, SERPINE1, SLC46A3, TNC) in JUNB KD and control conditions using the previously described RNA-sequencing data upon 100pM TGF$\beta$ stimulation (Fig 6A). As expected, JUNB KD specifically eliminated the late DDG amplification 720 min post-stimulation (FN1, SLC46A3, TNC) or had the more pronounced effect at this late time point (SERPINE1). In contrast, KD of the constitutive regulator SMAD2 diminished expression values of most DDGs already at earlier time points (SERPINE1, SLC46A3, TNC) and depletion of the early repressor SKI either had no effect on DDG expression (FN1, SLC46A3, TNC) or led to early up-regulation (SERPINE1) (Fig 6A). Having shown that JUNB promotes the late amplification of specific DDGs, we asked whether the same holds true on the complete set of 55 DDGs, which are not explained by the simple gene expression model. Indeed, our analysis confirms that JUNB KD had little effects on TGF$\beta$-induced fold changes at early time points, when leading to a homogenous and significant reduction for the 55 DDGs at the late 720 min time point (Fig 6B). In contrast, depletion of SMAD2 reduced global TGF$\beta$-induced DDG expression already at the earliest time points, reaching significance 360 and 720 min post-stimulation (Fig 6B).

To test whether the late amplification by JUNB is exclusive to the set of DDGs, we also analyzed the impact of SMAD2, JUNB, and SKI on other dynamic gene groups that could not be explained by the simple gene expression model (Figs 6B and S8A and B). For the delayed gene group, which is closely related to the DDGs, we again observed a specific and significant JUNB KD effect at the late time point, which however was small compared to the DDGs. For other gene groups, JUNB KD had no significant effect on TGF$\beta$-induced expression (biphasic and immediate early gene groups), or it barely reached significance (continuous gene group). Taken together, these data confirm the JUNB-mediated late amplification is most pronounced in DDGs when compared with other gene groups. In line with a specific late amplification by JUNB, the SMAD2 KD impacts all gene groups already at earlier time points (Figs 6B and S8A and B).

To further confirm the late DDG amplification by JUNB, we directly inferred time-dependent transcription rates of target genes in JUNB KD versus control conditions (Fig 6C). For inference we used a generic ODE describing mRNA expression as a function of transcription and mRNA degradation (see Equation (23)), and

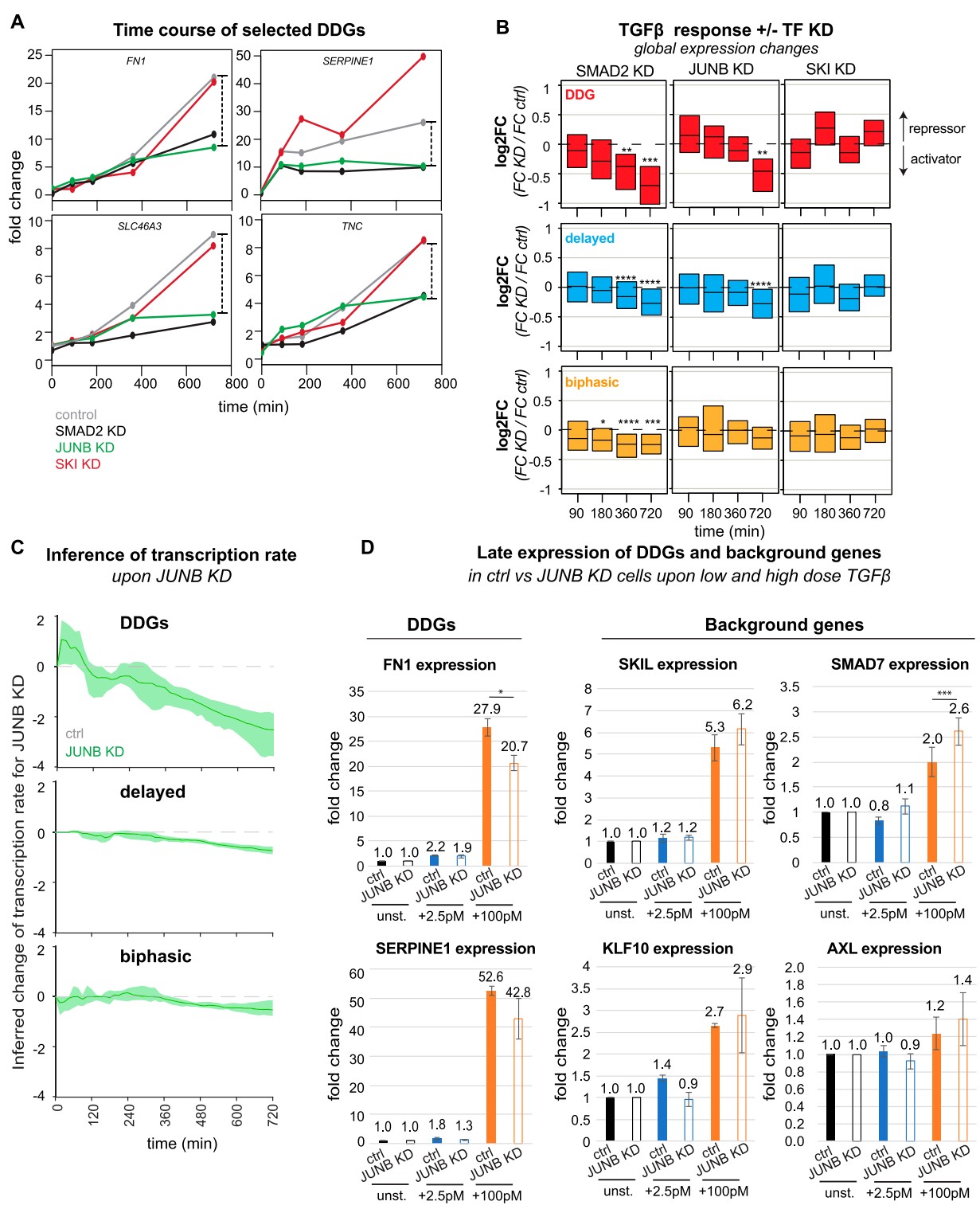

**Figure 6. JUNB reinforces TGFβ dose discrimination.**
**(A)** JUNB KD boosts late expression of selected DDGs upon high-dose TGFβ stimulation. Shown are time courses of DDGs (FN1, SERPINE1, SLC46A3, TNC), defined by late amplification for the strong TGFβ stimulus (Fig 2) in control (grey), SMAD2 KD (black), JUNB KD (green), and SKI KD (red) upon 100pM TGFβ stimulation (same RNA-sequencing data as in Fig 5B). Scale on the right indicates expression difference between control and JUNB KD condition at 720 min. **(B)** JUNB KD globally boosts late expression of DDGs (red, 43 genes), delayed target genes (blue, 290 genes), and biphasic target genes (yellow, 64 genes) rejected by the simple gene expression model (Fig 3). Boxes show changes of TGFβ-induced gene expression upon KD (log₂FC = [KD at t = x/KD at t = 0]/[control at t = x/control at t = 0]) as a distribution across all genes belonging to the indicated gene set (median with lower and upper quantile). SMAD2 KD significantly reduces TGFβ-induced gene expression changes at earlier time points compared with JUNB KD (t test). See Fig S8A and B for analysis of down-regulated genes and other gene groups (*P ≤ 0.05, **P ≤ 0.01, ***P ≤ 0.001, ****P ≤ 0.0001). **(C)** Time-

rearranged the formula to express the transcription rate as a function of target gene mRNA slope and expression level in the RNA-sequencing data (see Equation (24) and see the Materials and Methods section for details). As expected, the transcription rate of DDGs was strongly reduced upon JUNB KD, specifically at late time points (Fig 6C top). Delayed genes showed a similar drop in transcription, but with much milder amplitude (Fig 6C middle), and biphasic genes showed almost no (early time points) or minor (late time points) change upon JUNB KD (Fig 6C bottom). This regulatory pattern was observed in both, genes up-regulated by TGFβ (Fig 6C), and genes down-regulated by TGFβ (Fig S8C). Thus, our transcription-rate inference supports that JUNB is specific regulator for late DDG amplification.

To directly test whether JUNB mediates the decoding of TGFβ doses, we analyzed the expression of two EMT-relevant DDGs (SERPINE1 and FN1) whose gene promoters harbor JUNB binding sites (Table S8). Using quantitative PCR, we analyze expression in control and JUNB KD conditions upon stimulation with a low (2.5 pM) and high (100 pM) TGFβ dose for 1,440 min. In line with our expectation, we found that the JUNB KD specifically reduced SERPINE1 and FN1 expression in high-dose conditions, with lesser effects at the low dose (Fig 6D). In contrast, a background gene set, comprising rejected target genes with delayed expression kinetics but not classified as DDGs (SMAD7, SKIL, KLF10, and AXL), did not show a specific response at the high TGFβ dose and was generally insensitive to JUNB KD (Fig 6D). Taken together, our data suggest that JUNB is involved in the late amplification of DDGs and therefore may contribute to specific decision-making toward EMT for sufficiently strong TGFβ stimuli. The mechanism of late gene regulation by JUNB, however, remains elusive, as we detected only very weak JUNB protein up-regulation upon TGFβ treatment (Fig S9), suggesting that the late recruitment of JUNB to target gene promoters may, in addition, be controlled by unknown post-translational modifications or cooperating factors.

## Discussion

Understanding how ligand binding to cell surface receptors controls intracellular signaling, gene expression changes and ultimately cellular fate is critical for the development of therapeutic strategies targeting the TGFβ-pathway. In this work, we focused on specificity mechanisms in TGFβ-induced gene regulation, and characterized how TFs dynamically interact to drive changes in gene expression, and how this may be involved in shifting of SMAD signaling from a tumor suppressive to a tumor promoting function. Previous studies have demonstrated that the temporal dynamics of signaling

molecules, such as ERK, NFkB, and p53 determine phenotypic outcomes (Purvis & Lahav, 2013). Consequently, our objective was to explore the mechanisms responsible for decoding SMAD signaling dynamics into specific gene expression programs and cell fate decisions.

In our recent work, we reported that SMAD-induced gene expression exhibits little specificity within the first few hours after stimulation, as two TGFβ family ligands, GDF11 and TGFβ, induced very similar transcriptome changes across ligand doses and cellular stages, including quiescence and proliferation (Bohn et al, 2023). Previous studies reported distinct early and late gene expression programs in response to TGFβ treatment (Yang et al, 2003; Sundqvist et al, 2018). After this, we performed time-resolved RNA sequencing at a low and high-dose TGFβ stimulation to determine whether dose-specific gene expression programs emerge at later time points. Indeed, we found a subset of targets, termed DDGs, that was selectively regulated upon high-dose TGFβ stimulation after 12–24 h (Fig 2). Many of these DDGs show weak expression early after stimulation and are strongly amplified at late time points, suggesting a network of feedforward loops, in which SMAD-induced repressor dampen the DDGs at early time points, whereas activators such as JUNB specifically act at late time points and high doses to amplify the DDGs (Fig 7).

To classify target genes into those that simply respond to the SMAD signal and those that require additional input by late TFs, we devised a genome-wide modeling approach, in which the temporal evolution of each target gene is described taking into account the balance between transcription and mRNA degradation. Considering the experimentally measured SMAD input and fitting the model parameters to describe the RNA-sequencing time courses, we could explain 64% of the 4,823 differentially expressed target genes by this simple model (Fig 3). Similarly, gene expression models with an experimentally measured TF input disentangled input decoding in other cellular networks, such as the p53 and MAPK signaling pathways (Purvis et al, 2012; Uhlitz et al, 2017), and were also used to describe the dynamics of a small set of 12 TGFβ target genes in liver cells (Lucarelli et al, 2018). Likewise, similar dynamic models were used to infer mechanisms of biological regulation from multi-OMICS time course data (Peshkin et al, 2015; Becker et al, 2018). Interestingly, even though our model considered a flexible Hill-type sigmoidal function for transcriptional regulation, it failed to describe the dynamics of ~1,700 target genes, including most DDGs. This suggests that late dose-dependent amplification cannot be explained by switch-like promoter regulation but, in addition, requires input from dynamically changing TFs.

After this, we conducted KD experiments followed by RNA sequencing to elucidate the influence of TGFβ-induced TFs on target gene expression and found pronounced time-dependent

dependent target gene transcription rates (v(t)) of gene groups were inferred from RNA-sequencing data for JUNB KD and control cells. The transcription rate was separately inferred for each gene using the equation $v(t) = \tau \frac{dx}{dt} + x$, where x is the mRNA expression time course interpolated from RNA-sequencing data using constrained cubic splines (details see the Materials and Methods section). To quantify the effect of JUNB, the changes in the transcription rate upon JUNB KD ($\Delta v = v_{KD} - v_{control}$) were calculated for TGFβ–up-regulated DDGs (top), delayed (middle) and biphasic (bottom) genes. Lines: median of Δv in each across all genes in each group. Shades: bootstrapped confidence bands (2,000 bootstrap samples). Shown are the results for TGFβ–up-regulated genes; see Fig S8C for the same inference of TGFβ–down-regulated genes. **(D)** JUNB KD reduces dose discrimination of by DDGs. MCF10A control and JUNB KD cells were stimulated with 2.5 (blue) and 100 pM (orange) TGFβ for 1,440 min, and the expression of FN1 and SERPINE1 (n = 3) was assessed by qRT-PCR. In addition, the expression of a background gene set belonging to the delayed and biphasic kinetic groups, including SKIL, SMAD7, KLF10, and AXL (n = 3) was analyzed. SEM is shown.

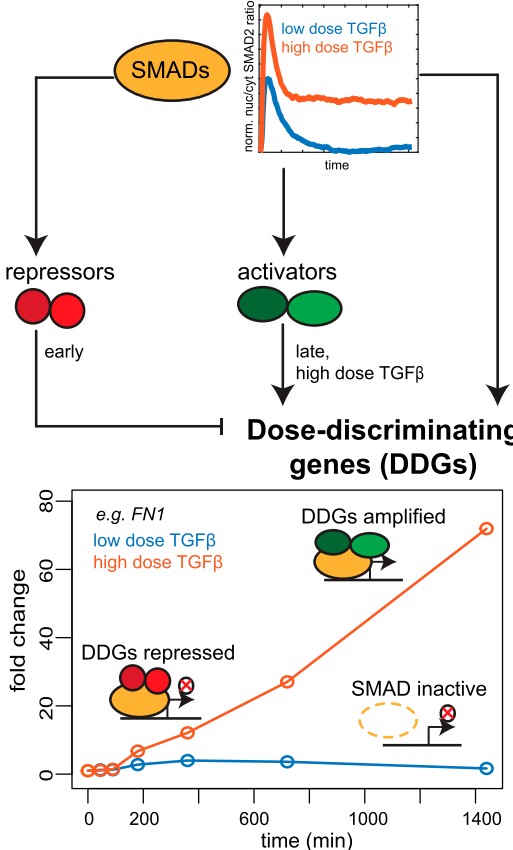

**Figure 7. Regulatory mechanisms of DDGs.**
The combination of multiple SMAD-dependent feedforward loops establishes dose discrimination at the level of target gene expression. SMAD TFs control expression of DDGs directly and indirectly by inducing the expression of transcriptional repressors (e.g., SKIL) and activators (e.g., JUNB). The repressors prevent the early up-regulation of DDGs, and the activators lead to their late amplification at the high TGF$\beta$ dose, thereby leading to dose discrimination.

regulatory effects: Among the tested TGF$\beta$-induced TFs, JUNB showed activator function, mainly acting at late time points, whereas the remaining co-regulators (SNAI1/2, SKI, SKIL, RUNX1) were classified as early repressors (Fig 5D). It is surprising that all of the negative regulators induced upon TGF$\beta$ treatment have a broad effect on TGF$\beta$ target genes. One possible explanation may be that many of them exert their effects by direct binding to SMAD proteins (Hanai et al, 1999; Luo et al, 1999; Stroschein et al, 1999; Vincent et al, 2009), acting additively or even cooperatively by modulating the recruitment of epigenetic co-repressors or co-activators to SMAD complexes. In line with our findings, SNAI2 has been identified as a key regulator of mammary epithelial cell identity (Phillips & Kuperwasser, 2014). Even though their time-dependent effects have not been characterized in detail, SKIL and SKI are known to be repressors of the TGF$\beta$-induced gene expression response (Tecalco-Cruz et al, 2018), which control EMT in non-small cell lung cancer (Yang et al, 2015) and metastasis in breast cancer cells (Le Scolan et al, 2008). Finally, JUNB is an important regulator of TGF$\beta$-induced invasive behavior and expression of EMT-related target genes (Sundqvist et al, 2018) that promotes EMT in MCF10A cells when overexpressed (Antón-García et al, 2023). Interestingly, the

regulating TFs identified in this work globally perturb the regulation of most if not all TGF$\beta$-dependent target genes, thereby exhibiting limited specificity in controlling the gene expression program. For instance, JUNB as an activator globally boosts both the up- and down-regulated genes, thereby acting as an amplifier of TGF$\beta$-dependent gene regulation irrespective of the direction of change. On the contrary, the repressors dampen the TGF$\beta$ response, simultaneously reducing most up- and down-regulated genes. Of note, some TFs, that is, JUNB, SNAI1, SNAI2 or RUNX1, do show signs of specific regulation, as their KD affects subsets of target genes much more than the rest (Fig S6, grey dots). Hence, these are strong candidates for controlling specific cell fates, for example, by predominantly controlling EMT genes, when having lesser effects on other targets. Notably, in terms of TGF$\beta$ dose discrimination, specificity can also arise by TFs homogeneously affecting all target genes at specific time points. For instance, the observed early repression and late activation of target genes by the tested TFs may ensure that a common global gene expression program is dampened for a transient SMAD input (low TGF$\beta$ dose), but amplified for a sustained SMAD signal (high TGF$\beta$ dose), thereby ensuring specific gene expression response at the high dose.

Currently, our findings concerning signal decoding are limited to MCF10A cells. However, given the similar dose-dependent SMAD signaling dynamics were measured in other cell lines, for example, in HaCaT or PE25 cells (Clarke et al, 2009; Zi et al, 2011), it is plausible that the feedforward loops discussed in this work are involved in signal decoding and EMT regulation in these systems as well. Accordingly, JUNB-dependent TGF$\beta$-induced EMT was reported in mouse mammary epithelial NMuMG cell line and the mouse kidney epithelial MCT cell line (Gervasi et al, 2012). However, in other cell types, other mechanisms of signal decoding may be more important, as JUNB did not show EMT-regulatory capacity in A549 lung cancer cells (Antón-García et al, 2023).

Complex target gene expression kinetics, for example, delayed, continuous, biphasic, immediate early, and inconsistent kinetics, could not be fitted by our SMAD input model and therefore require regulation by other dynamically changing TFs, as implemented in our extended FFL models. In similarity to previous work on EGF-induced transcriptional networks (Amit et al, 2007), we provide evidence for feedforward regulation in TGF$\beta$-dependent gene expression, as late target gene expression requires ongoing SMAD signaling (Figs 4C and S3D–F) and is particularly sensitive to the KD of induced TFs cooperating with SMAD proteins. In line with feedforward regulation by JUNB, it has been reported that JUNB redirects SMAD2 binding to new target gene promoters 16 h post-stimulation (Sundqvist et al, 2018), and that late target genes (6 h) are enriched for both SMAD and AP-1 TF binding motifs (Antón-García et al, 2023). Interestingly, in our data, both amplifying and repressing FFLs seem to mainly operate upon high-dose stimulation, as twice as many TFs were induced by a strong stimulus compared with low dose (Fig S3A). This suggests that FFLs mediate the late amplification of DDGs and selective EMT induction for strong TGF$\beta$ stimulation (Fig 7).

In summary, we used a systems biology approach, combining ODE modeling with time-resolved RNA sequencing and phenotypic assays to obtain a holistic understanding of how feedforward regulation and SMAD signaling dynamics shape target gene

expression. We demonstrate that the temporal dynamics of the SMAD signal and co-regulators of SMAD proteins shape the invasive behavior of cells. Therefore, our work points to potential intervention targets in malignancy and may serve as a blueprint for future quantitative studies linking signaling pathway dynamics to specific gene expression responses and cell fate decisions.

# Materials and Methods

## Cell culture

MCF10A cells were cultured in DMEM/F12 medium (#21331020; Thermo Fisher Scientific) supplemented with 100 ng/ml cholera toxin (#C8052; Sigma-Aldrich), 20 ng/ml EGF (#AF-100-15; Preprotech), 10 μg/ml insulin (#I9278; Sigma-Aldrich), 500 ng/ml hydrocortisone (#H0888; Sigma-Aldrich), GlutaMax (#35050038; Thermo Fisher Scientific), 5% horse serum (#16050122; Thermo Fisher Scientific), and 1x penicillin/streptomycin (#15140122; Thermo Fisher Scientific). To culture H2B and SMAD2-tagged MCF10A cells, 200 μg/ml Geneticin (#10131035; Thermo Fisher Scientific) and 20–50 μg/ml Hygromycin (#10687010; Thermo Fisher Scientific) were added to the medium. Cells were trypsinized for ~15 min with 0.05% Trypsin/EDTA (#15400054; Thermo Fisher Scientific).

## Ligand and inhibitor treatment

Lyophilized TGFβ-1 derived from CHO-cells (#100-21C; Prepotech) was reconstituted in 0.1% BSA and 4 mM HCl. MCF10A cells were cultured 2 d before stimulation with different doses of TGFβ. To stop TGFβ-signaling a TGFβ-receptor inhibitor (SB431542) (#616461; Merck) blocking the kinase activity of TGFβ –receptor-II was applied 90 min or 300 min post-stimulation with a final concentration of 10 μM (Figs 4 and S3). In long-term stimulation experiments, old medium was aspirated and cells were re-stimulated every 48 h with TGFβ-1.

## SMAD2 nuclear translocation data

SMAD2 nuclear translocation data for 351 and 378 single cells upon 2.5- and 100 pM-TGFβ stimulation, respectively, with a sampling rate of every 5 min over a period of 24 h, were originally published in Strasen et al (2018); Strasen et al (2018) (Fig 1). Here, we only considered the population-median at each time point across all individual cells, therefore neglecting cellular heterogeneity. Furthermore, the original data, given as a nuclear-to-cytoplasmic ratio of the SMAD2 protein, were background-subtracted to provide a zero-baseline input for mathematical modeling (Fig 3).

## siRNA-mediated knockdown

siRNA-mediated KD was performed using Dharmacon ON-TARGETplus Human siRNA SMART pools containing a selection of four different siRNA sequences targeting the gene of interest (SMAD2: #L-003561-00-0005, SMAD3: #L-020067-00-0005, SNAI1: #L-010847-00-0005, SNAI2: #L-017386-00-0005, RUNX1: #L-003926-00-0005, SKIL: # L-010535-00-0005, SKI: #L-003927-00-0005, JUNB: #L-003269-00-0005, JUN: #L-003268-00-

0005, KLF10: #L-006566-00-0005, ATF3: #L-008663-00-0005). A pool of non-targeting siRNAs was used as negative control (#D-001810-10; Dharmacon). Reverse transfection with RNAiMAX (#13778075; Lipofectamine RNAiMAX) was performed at 10 nM final siRNA concentration. For $2 × 10^6$ cells in 10 ml cell culture medium lacking any antibiotics, 500 μl Opti-MEM (#1985062), 5 μl siRNA (stock concentration: 20 μM) and 10 μl RNAiMAX were pre-incubated for 20 min at RT before transfection. For TGFβ stimulation experiment, MCF10A cells were stimulated with 100pM TGFβ 24 h post-KD generation and harvested at the time of stimulation (0 h) and later time points (90/180/360/720 min post-stimulation) (Figs 5, S5, and S6). To conduct phenotypic analysis on KDs (see the flow-cytometry analysis section), the siRNA KD was renewed every 48 h (Figs 5E and S7).

## Bulk RNA sequencing and qRT-PCR

RNA was isolated using the RNeasy Plus RNA-isolation kit (#74136; QIAGEN) according manufactures instructions. Quantification of RNA concentrations and purity were determined with the NanoDrop. RNA quality was assessed and analyzed using RNA Screen Tapes and specific sample buffer (#RNA-Screen Tapes #5067–5576, RNA Screen Tape sample buffer #5067–5577, RNA Screen Tape Ladder #5067–5578) in the Agilent TapeStation device. Calculated RINe values > 6 with flat baseline were strived for RNA-sequencing experiments. Library preparation and bulk RNA-sequencing experiments were performed in biological triplicates by Novogene (Figs 4–6) using poly-A-enrichment and paired-end-sequencing of 150 bp a NovaSeq 6000 instrument with collecting 20 mio reads.

Library preparation of time course bulk RNA-sequencing experiments upon low and high dose of TGFβ and initial data processing were performed in biological triplicates by the IMB Genomics Core Facility (Mainz) (Figs 1 and 2). After poly-A-enrichment, transcripts were sequenced by single-end-sequencing of 75 bp on the NextSeq 500 sequencing platform.

Quantitative real-time reverse-transcription PCR (qRT-PCR) was performed in a two-step process. First, the SuperScript Reverse Transcriptase kit was used to generate cDNA according the manufactures instruction (#18064014; Thermo Fisher Scientific) using Olig(dT)-primer for mRNA specificity. Specific forward and reverse primers for target genes of interest were designed and purchased from IDT (SMAD2; FWD: 5′ GGGTTTTGAAGCCGTCTATCAGC 3′, REV: 5′ CCAACCACTGTAGAGGTCCATTC 3′, SMAD3; FWD: 5′ CATCGAGCCCCA-GAGCAATA 3′, REV: 5′ GTAACTGGCTGCAGGTCCAA 3′, SNAI1; FWD: 5′ GCCTAGCGAGTGGTTCTTCT 3′, REV: 5′ TGCTGGAAGGTAAACTCTGGATT 3′, SNAI2; FWD: 5′ TTCAACGCCTCCAAAAAGCC 3′, REV: 5′ AGAGA-TACGGGGAAATAATCACTGT 3′, RUNX1; FWD: 5′ CGTGGTCCTACGATCAGTCC 3′, REV: 5′ GTCGGGTGCCGTTGAGA 3′, SKIL; FWD: 5′ GGCTGAA-TATGCAGGACAG 3′, REV: 5′ TGAGTTCATCTTGGAGTTCTTG 3′, SKI; FWD: 5′ TTCCGAAAAGGACAAGCCGT 3′, REV: 5′ ACAGCCCAGGCTCTTATTGG 3′, JUNB; FWD: 5′ CCTGGACGATCTGCACAAGA 3′, REV: 5′ GTAGCTGCTGAGGTTGGTGT 3′, JUN; FWD: 5′ CCTGGACGATCTGCACAAGA 3′, REV: 5′ GTAGCTGCT-GAGGTTGGTGT 3′, KLF10; FWD: 5′ CGAGGACGCACACAGGAGAA 3′, REV: 5′ GTCACTCCTCATGAACCGCC 3′, ATF3; FWD: 5′ TCCATCACAAAAGCCGAGGTAG 3′, REV: 5′ CTGCAGGCACTCCGTCTTC 3′, SERPINE1: FWD: 5′ GGCTGACTT-CACGAGTCTTTCA 3′, REV: 5′ ATGCGGGCTGAGACTATGACA 3′, FN1: FWD: 5′ GTGTGATCCCGTCGACCAAT 3′, REV: 5′ CGACAGGACCACTTG AGCTT 3′, SMAD7: FWD: 5′ ACCCGATGGATTTTCTCAAACC 3′, REV:

5′ GCCAGATAATTCGTTCCCCCT 3′, AXL: FWD: 5′ ACTCTGGGAGAGGGAGAGTT 3′, REV: 5′ GAGCCTCATGACGTTGGGAT 3′) and used at final concentrations of 240 nM. Second, for qRT-PCR the Power SYBR Green PCR Master Mix (#4367659; Thermo Fisher Scientific) was used at 1x final concentration having SYBR Green I Dye, AmpliTaq Gold DNA Polymerase, dNTPs and internal passive reference dye (ROX), included. The qRT-PCR was performed on a QuantStudio 5 Real-Time-PCR-System (Figs 6C, S5A, and S7A).

## Flow-cytometry analysis

The expression levels of vimentin and E-cadherin were assessed at the protein level by flow cytometry (Fig 2). MCF10A cells were treated for 5 or 8 d with TGF-$\beta$1 (1/5/50/250 pM or 100 pM), with subsequent restimulation every 48 h. To assess the impact of co-factors on EMT initiation, cells were first transfected and then stimulated 24 h later for 8 d with 100 pM TGF-$\beta$1 with subsequent restimulation every 48 h (Figs 5 and S7). Cells were harvested by TrypLE (#12604013; Thermo Fisher Scientific). Before fixation, cells were stained with LIVE/DEAD Fixable Dead Cell Stain for 30 min at 4°C (#L34975; Thermo Fisher Scientific) to differentiate between living and dead cells. For cell fixation and permeabilization, cells were treated with 4% PFA in DPBS for 15 min, and 0.1% Triton in FACS media (DPBS + 5% FHS) for 15 min at 4°C. Cells were then incubated with anti-CD324 (E-cadherin) antibody (#14472, clone 4A2, monoclonal mouse Ab, 1:150; Cell signaling) and anti-vimentin antibody (#5741, clone D21H3, monoclonal rabbit Ab, 1:150; Cell signaling) for 1 h. After blocking in FACS media for 10 min, cells were incubated with secondary antibodies Alexa Fluor 647 goat anti-mouse IgG (H+L) (#2369432, 1:1,000; Thermo Fisher Scientific) and Alexa Fluor 488 goat anti-rabbit IgG (H+L) (#2382186, 1:250; Thermo Fisher Scientific) for 45 min. All antibody incubations were performed in FACS media. Cells were washed between live-dead-staining and fixation with DPBS and during antibody staining with FACS media for two times and analyzed by MACSQuant10 flow-cytometer. Flow-cytometry data were analyzed by FlowJo v10.8.1, quantifying FITC-A SSC-A (vimentin) or APC-A SSC-A (E-cadherin) in the cell population.

## Bioinformatics analysis of bulk RNA-sequencing data

Basic bioinformatics analysis from raw sequencing files to DESeq2 was performed by the bioinformatics core facility at the IMB Mainz (*time course data*) and Novogene (*KD data*). Raw sequence reads were assessed with FastQC (v. 0.11.5) (Andrews, 2010) and aligned to the human reference genome GRCh38.p12 (GTF annotation file from Gencode human release 25) using STAT aligner (v. 2.5.2b) or Hisat2 aligner (v. 2.0.5) (Dobin et al, 2013; Kim et al, 2015) for the KD data, respectively. Mapped data were summarized on the gene level using the featureCounts software (v.1.5.0-p3) (Liao et al, 2014). Further data evaluation, normalization and pairwise differential expression analyses were carried out in R using the Bioconductor package DESeq2 (*v.1.18.1, time course data, v.1.20.0, KD data*) (Love et al, 2014). To investigate the number of genes, significant differentially regulated transcripts were filtered for protein coding genes and RPKM/FPKM > 0.5. Differentially expressed genes were isolated based on FDR < 0.01 and absolute fold change ≥ 1.5. Phenotypic specific gene groups were selected from different sources (see Tables S1, S2, and S3); EMT genes: curated general EMT gene list (Zhao et al, 2015), and the TGF$\beta$-EMT gene signature (Gordian et al, 2019), cell cycle genes: https://www.gsea-msigdb.org/gsea/msigdb/cards/KEGG_CELL_CYCLE.

## Implementation of a minimal gene expression model

ODE models were used to describe the effect of SMAD2 on the dynamics of target gene expression. As an input for the model, we used background-subtracted microscopy data (see the SMAD2 nuclear translocation data section) with spline interpolation between 5 min sampling intervals. Transcriptional regulation by SMAD was modeled using Hill function and two possible cases were considered-activation and inhibition of target genes by SMAD. The ODE for the activation model reads.

$$\frac{dx}{dt} = \text{syn} + m \cdot \frac{SMAD^h}{SMAD^h + k^h} - \beta_0 \cdot x \tag{3}$$

where $x$ is the concentration of target gene mRNA, whereas syn and $\beta_0$ are the basal synthesis and degradation rates, respectively. In the Hill term, SMAD is the nuclear concentration of SMAD, $m$ = is the maximal SMAD-induced transcriptional rate, $h$ the Hill coefficient, and $k$ the half-saturation point. To directly compare our model with RNA-sequencing data, which quantifies the fold change rather than concentration of mRNA in response to TGF$\beta$ stimulation, we divided both sides of Equation (3) by the steady-state mRNA level without TGF$\beta$ stimulation, and the steady-state solution can be derived by the following.

$$\text{syn} + m \cdot \frac{SMAD^h}{SMAD^h + k^h} - \beta_0 \cdot x_0 = 0 \tag{4}$$

without TGF$\beta$ stimulation, SMAD = 0, so we have,

$$\text{syn} - \beta_0 x_0 = 0 \tag{5}$$

$$x_0 = \frac{\text{syn}}{\beta_0} \tag{6}$$

We then divided both sides of Equation (3) by $x_0$:

$$\frac{dy}{dt} = \frac{d}{dt}\left(\frac{x}{x_0}\right) = \frac{\text{syn}}{x_0} + \frac{m}{x_0}\frac{SMAD^h}{SMAD^h + k^h} - \beta_0\frac{x}{x_0} \tag{7}$$

where the steady-state expression $x_0 = \frac{\text{syn}}{\beta_0}$ can be used to substitute $\frac{\text{syn}}{x0} = \beta_0$ and $\frac{m}{x_0}$ replaced with $k_{syn}$

$$\frac{dy}{dt} = \beta_0 + k_{syn}\frac{SMAD^h}{SMAD^h + k^h} - \beta_0 y \tag{8}$$

We can then use the above ODE solution to directly compare with the time-resolved RNA-sequencing data. The same principle was used for the inhibition model (see supplementary methods). We fit both activation and inhibition models independently to each gene by minimizing the $\chi^2$ (Sheskin, 2003) value defined as,

$$\chi^2 = \sum_{i=1}^{N}\left[\frac{y_{\text{data}}(t_i) - y_{\text{model}}(t_i)}{\Delta r \cdot y_{\text{data}}(t_i)}\right]^2 \tag{9}$$

where $y_{data}(t_i)$ and $y_{model}(t_i)$ are the measured and modeled mRNA fold changes at time point $t_i$, respectively. The RNA-sequencing data were collected from 12 time points in total (6 for high dose and 6 for low dose of TGFβ) for each gene, that is, $N = 12$. The open-source Matlab package Data2Dynamics (D2D) was used for model fitting (Raue et al, 2013, 2015). To avoid local minima, the Latin hypercube sampling (n = 500) was used to sufficiently scan the initial parameter space. To determine the relative error, we calculate the slope of a scatter plot, plotting mean RPKMs (a) versus SD (b) (see Fig S10A). We make use of this relative error (r) and use the Gaussian error propagation (see supplementary methods).

$$r = \frac{a}{b} \tag{10}$$

Using the calculated slope of the linear error model on RPKMs (m = 0.082) (see Fig S10A), the Gaussian error propagation equation results in a relative error of 0.11; $\Delta r = \sqrt{2} \cdot 0.082 = 0.11$ For both the activation and inhibition models, we used a $\chi^2$ goodness-of-fit test (Cedersund & Roll, 2009) to assess their agreement with the experimental data. The decision to accept or reject a model was determined by comparing the $\chi^2$ value from the best fit with the critical $\chi^2$ value given by the $\chi^2$ distribution (with corresponding degrees of freedom) at 95% confidence level. The degrees of freedom are 8 (12 data points minus 4 free parameters) for the activation model and 9 (12 data points minus 3 free parameters) for the inhibition model. The critical $\chi^2$ values were established as 15.5 for the activation model and 16.9 for the inhibition model. If the fitted $\chi^2$ is lower than the corresponding critical $\chi^2$ value, the model is accepted; otherwise, the model is rejected. In case that both activation and inhibition models were accepted by the $\chi^2$ test for a gene, we used the Akaike information criteria (AIC) (Akaike, 1974) for further model discrimination,

$$AIC = -2\log L + 2p \sim \chi^2 + 2p \tag{11}$$

where $L$ is the likelihood and $p$ the number of free parameters. Because here we assume normally distributed errors, the $\chi^2$ value and $-2\log L$ only differ by a constant value and therefore can be used to calculate AIC for different models. The model with the lower AIC was selected. Based on these criteria, we were able to describe 19% of target genes by the activation model and 45% by the inhibition model. Both models were rejected for the remaining 36% (1,744 of 4,823) target genes (Fig 3).

## Model extension by feedforward loops

To explain the genes rejected by the simple activation and inhibition models (see above), we implemented extended models taking into account FFLs (Mangan & Alon, 2003). We introduced an unknown regulator, named TF, which is activated directly by SMAD. TF regulates the target gene (TG), as either an activator or a repressor, jointly with SMAD via AND- or OR-gate logic. This generates in total eight different models (Figs 4D and S4G). There are two equations for each of the eight model variants. The first

equation describes the activation of TF and is similar to the simple activation model (1).

$$\frac{dTF}{dt} = \beta_0 + r \cdot \frac{SMAD^h}{SMAD^h + k^h} - \beta_0 TF \tag{12}$$

The second equation differs among the eight models depending on activating or repressor role of TF and SMAD. In addition, the transcription rate changes if we assume an AND-gate = f(SMAD) × f(TF) or OR-gate logic f(SMAD) + f(TF) – f(SMAD) × f(TF) (Rogers et al, 2022) (see supplementary methods for initial FFL equations and their steady state). By dividing the equations by the steady state (Supplementary Methods), we obtain the following equations for each model:

Model 1: AND-gate: SMAD activator, TF activator

$$\frac{dTG}{dt} = b_{tg} + rs \cdot \frac{SMAD^{hs_1} \cdot TF^{hp}}{\left(SMAD^{hs_1} + ks_1^{hs_1}\right) \cdot \left(kp^{hp} + TF^{hp}\right)} - b_{tg} \cdot TG \tag{13}$$

Model 2: OR-gate: SMAD activator, TF activator

$$\frac{dTG}{dt} = \frac{syn_{tg} + rp \cdot \frac{TF^{hp}}{kp^{hp} + TF^{hp}} + rs \cdot \frac{SMAD^{hs_1}}{\left(SMAD^{hs_1} + ks_1^{hs_1}\right)} - rp \cdot rs \cdot \frac{SMAD^{hs_1} \cdot TF^{hp}}{\left(SMAD^{hs_1} + ks_1^{hs_1}\right) \cdot \left(kp^{hp} + TF^{hp}\right)}}{\frac{syn_{tg} + rp \cdot \frac{1}{kp^{hp} + 1}}{b_0}} - b_{tg} \cdot TG \tag{14}$$

Model 3: AND-gate: SMAD repressor, TF activator

$$\frac{dTG}{dt} = \frac{syn_{tg} + rp \cdot \frac{ks_1^{hs_1} \cdot TF^{hp}}{\left(SMAD^{hs_1} + ks_1^{hs_1}\right) \cdot \left(kp^{hp} + TF^{hp}\right)}}{\frac{syn_{tg} + rp \cdot \frac{1}{kp^{hp} + 1}}{b_{tg}}} - b_{tg} \cdot TG \tag{15}$$

Model 4: OR-gate: SMAD repressor, TF activator

$$\frac{dTG}{dt} = b_{tg} \cdot \frac{ks_1^{hs_1}}{SMAD^{hs_1} + ks_1^{hs_1}} + rp \cdot \frac{TF^{hp}}{\left(kp^{hp} + TF^{hp}\right)}$$
$$- rp \cdot \frac{ks_1^{hs_1} \cdot TF^{hp}}{\left(SMAD^{hs_1} + ks_1^{hs_1}\right) \cdot \left(kp^{hp} + TF^{hp}\right)} - b_{tg} \cdot TG \tag{16}$$

Model 5: AND-gate: SMAD activator, TF repressor

$$\frac{dTG}{dt} = b_{tg} + rs \cdot \frac{SMAD^{hs_1} \cdot kp^{hp}}{\left(SMAD^{hs_1} + ks_1^{hs_1}\right) \cdot \left(kp^{hp} + TF^{hp}\right)} - b_{tg} \cdot TG \tag{17}$$

Model 6: OR-gate: SMAD activator, TF repressor

$$\frac{dTG}{dt} = \frac{syn_{tg} \cdot \frac{kp^{hp}}{kp^{hp} + TF^{hp}} + rs \cdot \frac{SMAD^{hs_1}}{\left(SMAD^{hs_1} + ks_1^{hs_1}\right)} - rs \cdot \frac{SMAD^{hs_1} \cdot kp^{hp}}{\left(SMAD^{hs_1} + ks_1^{hs_1}\right) \cdot \left(kp^{hp} + TF^{hp}\right)}}{\frac{syn_{tg} \cdot kp^{hp}}{b_{tg} \cdot \left(kp^{hp} + 1\right)}} - b_{tg} \cdot TG \tag{18}$$

Model 7: AND-gate: SMAD repressor, TF repressor

$$\frac{dTG}{dt} = \frac{syn_{tg} \cdot \frac{kp^{hp} \cdot ks_1^{hs_1}}{\left(SMAD^{hs_1} + ks_1^{hs_1}\right) \cdot \left(kp^{hp} + TF^{hp}\right)}}{\frac{syn_{tg} \cdot kp^{hp}}{b_{tg} \cdot \left(kp^{hp} + 1\right)}} - b_{tg} \cdot TG \tag{19}$$

Model 8: OR-gate: SMAD repressor, TF repressor

$$\frac{dTG}{dt} = b_{tg} \cdot \left( \frac{ks_1{}^{hs_1}}{SMAD^{hs_1} + ks_1{}^{hs_1}} + \frac{kp^{hp}}{kp^{hp} + TF^{hp}} \right)$$
$$- b_{tg} \cdot \frac{kp^{hp} \cdot ks_1{}^{hs_1}}{\left( SMAD^{hs_1} + ks_1{}^{hs_1} \right) \cdot \left( kp^{hp} + TF^{hp} \right)} - b_{tg} \cdot TG \qquad (20)$$

where the parameters are defined as follows:

$Syn_{tg}$ = basal target gene synthesis rate.
rs = maximal rate of SMAD-induced target gene (TG) transcription.
rp = maximal rate of TF-induced target gene transcription.
hs1 = hill coefficient of SMAD-induced target gene transcription.
ks1 = half-saturation point of SMAD regulation.
hp = hill coefficient of TF-induced target gene transcription.
kp = half-saturation point of TF regulation.
$b_{tg}$ = degradation rate of target gene mRNA.

The 8 FFL models were fitted to individual rejected genes using the same method as applied earlier for the simple model. As shown above, the model Equations (13), (14), (15), (16), (17), (18), (19), and (20) for the FFL possess many parameters (up to 12). To obtain the effective degrees of freedom (DOF) for the $\chi^2$ test, we applied a simulation-based method proposed by Kreutz (Kreutz, 2020) (see Fig S10B). To determine the DOF for each of the 8 FFL model, we generated synthetic data from the best fit and gain a model-intrinsic $\chi^2$ by refitting.

First, we selected 14 rejected genes (by simple models) that represent different dynamic groups (Fig 4). Then, we fit 8 FFL models to each of them and selected the best model using AIC. For each gene, we added normally distributed noise with 11% SD to the time trajectories of the best fit, generating 100 synthetic data sets (Fig S10B). We then refit the model to the 100 samples and calculated the $\chi^2$. The $\chi^2$ values before refit is quantified as:

$$\chi_r{}^2 = \sum_{i=1}^{N} \left[ \frac{y_{\text{first fit of the genes}}(t_i) - y_{\text{randomly generated simulation}}(t_i)}{\Delta r \cdot y_{\text{first fit of the genes}}(t_i)} \right]^2 \quad (21)$$

and after refit:

$$\chi_{fr}{}^2 = \sum_{i=1}^{N} \left[ \frac{y_{\text{randomly generated simulation}}(t_i) - y_{\text{re-fitted final result}}(t_i)}{\Delta r \cdot y_{\text{randomly generated simulations}}(t_i)} \right]^2 \quad (22)$$

We calculated the model-intrinsic degrees of freedom following the formula $DOF = \chi_r{}^2 - \chi_{fr}{}^2 = 12 - 6 = 6$ (Fig S10C). Based on this effective degrees of freedom, we performed model rejection/selection on all 1,744 genes (rejected by the simple models) using $\chi^2$ test or/and AIC.

## Classification of rejected genes

To investigate why the simple model is not able to explain 1,744/4,823 differentially expressed genes, we analyzed the gene expression kinetics shared by the rejected genes and are rarely found in the described genes. For that, we have grouped rejected genes into five subgroups (Fig 4) "delayed expression kinetics,"

"inconsistent expression kinetics," "biphasic expression kinetics," "immediate early expression kinetics," and "continuous expression kinetics," by using different filter settings as indicated in Table S4.

## Identification of candidate TFs mediating feedforward regulation

We have used an earlier published (https://www.sciencedirect.com/science/article/pii/S0092867418301065) list of described TFs (Lambert et al, 2018). Across all early time points (<360 min) 99 TFs are differentially expressed upon low-dose stimulation and 133 TFs are differentially expressed upon high-dose stimulation. We further isolated TFs, which are following the SMAD dynamics and are therefore SMAD-induced TF and share differentially expression between 45–90 min post-stimulation. To further narrow the list of TFs, we filtered co-factors already described in literature to interact with SMADs. This result in a list of six co-factors (SNAI2, JUN, JUNB, SMAD7, KLF10, SKIL) differentially expressed upon low-dose application (Fig 5). The same six co-factors and four additional ones (RUNX1, SKI, E2F5, KDM6B) are differentially expressed upon high-dose application (Fig 5). We have chosen only the up-regulated TFs to focus on (SNAI2, RUNX1, SKIL, SKI, JUNB, JUN, KLF10) and added ATF3, SNAI1, as well as SMAD2 and SMAD3 as positive controls.

## Assessing impact of co-factor KD on target gene expression (slope quantification)

To divide SMAD co-factors into activators and repressors, we quantified the slope of scatter plots with gene expression values of the non-targeting control plotted on the x-axis against gene expression values of the KD (KD) condition shown on the y-axis (Figs 5 and S6). With this analysis we quantified whether the gene expression is on average, stronger or weaker upon KD compared with control. For further analysis, we only selected KDs with significantly changed slopes ≠ 1 ($t$ test, alpha = 0.05). Furthermore, we identified the variance among the three biological replicates by plotting their FPKM-values in scatter plots against each other (Table S6).

## Inference of transcription rate upon JUNB KD

To infer the time-dependent transcription rate without prior knowledge of molecular interactions among SMAD and co-factors, such as JUNB, we started with a generic ODE describing the kinetics of fold change in mRNA expression ($x$),

$$\frac{dx}{dt} = \frac{1}{\tau}[v(t) - x] \qquad (23)$$

where $v(t)$ is the TGF$\beta$-stimulated transcription rate relative to the basal transcription rate and thus $v(0) = 1$. $t$ is the mRNA lifetime. By rearranging the Equation (23), we obtained the transcription rate $v(t)$,

$$v(t) = \tau \frac{dx}{dt} + x \qquad (24)$$

Equation (24) suggests that the transcription rate $v(t)$ can be directly inferred from the time-resolved RNA-sequencing data. To overcome that the calculation of the time derivative ($dx/dt$) is

hindered by sparse time points, we used piecewise cubic-spline interpolation to yield interpolated mRNA trajectories with higher time resolution using a 15 min sampling interval. To avoid over flexibility of the naïve cubic splines, we used the modified Akima interpolation method implemented in Matlab (makima). We then calculated numerically the discrete time derivative ($\Delta x/\Delta t$) for each interpolated time point. For the value of mRNA lifetime $\tau$, we used a fixed value of 9 h for all mRNAs, corresponding the median mRNA lifetime in mammalian cells (Schwanhäusser et al, 2011). Notably, according to the Equation (24), the lifetime is just a scaling factor of the time derivative, mainly controlling the amplitude of the inferred transcription rate $v(t)$ with little effects on its temporal shape. To quantitatively understand the regulatory role of JUNB, we separately performed the transcription-rate inference for each gene belonging to the DDG, delayed, and biphasic gene groups and calculated the difference of transcription rates between JUNB KD and control conditions ($\Delta v(t) = v_{KD} - v_{control}$). For each group, we then calculated the median transcription-rate difference $\Delta v(t)$ at each time point, separately considering the up-regulated (Fig 6C) and down-regulated (Fig S8C) genes (defined based on their direction of change at the final time point (720 min) in the control condition).

# Supporting Information Text

### Methods

#### *Implementation of a minimal gene expression model: inhibition model*

The same principle used for the activation model (see the Materials and Methods section) was used for the inhibition model. The equation for the inhibition model is

$$\frac{dx}{dt} = \text{syn} \cdot \frac{k^h}{SMAD^h + k^h} - \beta_0 \cdot x \tag{25}$$

When t = 0 and SMAD = 0, and therefore $\frac{k^h}{SMAD^h + k^h} = 1$, the steady state is defined by

$$0 = \text{syn} \cdot 1 - \beta_0 \cdot x0 \tag{26}$$

$$X_0 = \frac{syn}{\beta_0} \tag{27}$$

We define the mRNA expression fold change $y \equiv \frac{x}{x_0}$. The respective ODE can be derived from Equation (25):

$$\frac{dy}{dt} = \frac{d}{dt}\left(\frac{x}{x_0}\right) = \frac{syn}{x_0} \cdot \frac{k^h}{SMAD^h + k^h} - \beta_0 \frac{x}{x_0} \tag{28}$$

where the steady-state expression $x_0 = \frac{syn}{\beta_0}$ can be used to substitute $\beta_0 = \frac{syn}{x_0}$.

$$\frac{dy}{dt} = \beta_0 \cdot \frac{k^h}{SMAD^h + k^h} - \beta_0 \cdot y \tag{29}$$

Or equivalently,

$$\frac{dy}{dt} = \beta_0 - \beta_0 \cdot \frac{SMAD^h}{SMAD^h + k^h} - \beta_0 \cdot y \tag{30}$$

To calculate the error, we use the Gaussian Error propagation:

$$\Delta F = \sqrt{\sum_x \left(\frac{\partial F}{\partial x}\right)^2 \Delta x^2} \tag{31}$$

where F is the fold change, r = a/b, and x = (a,b)

$$\Delta r = \sqrt{\left(\frac{\partial r}{\partial a}\right)^2 \cdot \Delta a^2 + \left(\frac{\partial r}{\partial b}\right)^2 \cdot \Delta b}$$

$$\frac{\partial r}{\partial a} = \frac{1}{b}; \frac{\partial r}{\partial b} = -\frac{a}{b^2} \tag{32}$$

$$\Delta r = \sqrt{\frac{\Delta a^2}{b^2} + \frac{a^2 \cdot \Delta b^2}{b^4}} = \sqrt{\frac{\Delta a^2 \cdot b^2}{b^4} + \frac{a^2 \Delta b^2}{b^4}} \tag{33}$$

Using the slope m of the linear error model on RPKMs which is defined by the absolute values a and b instead of the relative r, we substitute $\Delta a = m \cdot a$; and $\Delta b = m \cdot b$ and receive.

$$\Delta r = \sqrt{\frac{m^2 \cdot a^2 \cdot b^2}{b^4} + \frac{a^2 \cdot m^2 \cdot b^2}{b^4}} = \sqrt{2 \cdot m^2 \cdot \left(\frac{a}{b}\right)^2} \tag{34}$$

The following equation is used to calculate the relative error based on the calculated slope.

$$\Delta r = \sqrt{2 \cdot m^2 \cdot r^2} \tag{35}$$

### Model extension by feedforward loops

The equations for the eight extended models considering FFL regulation in an AND- or OR-gate logic (SF 3, G) are listed below without normalization by steady state:

Model 1: AND-gate: SMAD activator, TF activator.

$$\frac{dTG}{dt} = syn_{tg} + rs \cdot \frac{SMAD^{hs_1} \cdot TF^{hp}}{\left(SMAD^{hs_1} + ks_1^{hs_1}\right) \cdot \left(kp^{hp} + TF^{hp}\right)} - b_{tg} \cdot TG \tag{36}$$

Model 2: OR-gate: SMAD activator, TF activator.

$$\frac{dTG}{dt} = syn_{tg} + rp \cdot \frac{TF^{hp}}{kp^{hp} + TF^{hp}} + rs \cdot \frac{SMAD^{hs_1}}{\left(SMAD^{hs_1} + ks_1^{hs_1}\right)}$$
$$- rp \cdot rs \cdot \frac{SMAD^{hs_1} \cdot TF^{hp}}{\left(SMAD^{hs_1} + ks_1^{hs_1}\right) \cdot \left(kp^{hp} + TF^{hp}\right)} - b_{tg} \cdot TG \tag{37}$$

Model 3: AND-gate: SMAD repressor, TF activator.

$$\frac{dTG}{dt} = syn_{tg} + rp \cdot \frac{ks_1^{hs_1} \cdot TF^{hp}}{\left(SMAD^{hs_1} + ks_1^{hs_1}\right) \cdot \left(kp^{hp} + TF^{hp}\right)} - b_{tg} \cdot TG \tag{38}$$

Model 4: OR-gate: SMAD repressor, TF activator.

$$\frac{d\text{TG}}{dt} = syn_{tg} \cdot \frac{ks_1^{hs_1}}{\text{SMAD}^{hs_1} + ks_1^{hs_1}} + rp \cdot \frac{TF^{hp}}{\left(kp^{hp} + TF^{hp}\right)}$$
$$- rp \cdot \frac{ks_1^{hs_1} \cdot TF^{hp}}{\left(\text{SMAD}^{hs_1} + ks_1^{hs_1}\right) \cdot \left(kp^{hp} + TF^{hp}\right)} - b_{tg} \cdot TG \quad (39)$$

Model 5: AND-gate: SMAD activator, TF repressor.

$$\frac{d\text{TG}}{dt} = syn_{tg} + rs \cdot \frac{\text{SMAD}^{hs_1} \cdot kp^{hp}}{\left(\text{SMAD}^{hs_1} + ks_1^{hs_1}\right) \cdot \left(kp^{hp} + TF^{hp}\right)} - b_{tg} \cdot TG \quad (40)$$

Model 6: OR-gate: SMAD activator, TF repressor.

$$\frac{d\text{TG}}{dt} = syn_{tg} \cdot \frac{kp^{hp}}{kp^{hp} + TF^{hp}} + rs \cdot \frac{\text{SMAD}^{hs_1}}{\left(\text{SMAD}^{hs_1} + ks_1^{hs_1}\right)}$$
$$- rs \cdot \frac{\text{SMAD}^{hs_1} \cdot kp^{hp}}{\left(\text{SMAD}^{hs_1} + ks_1^{hs_1}\right) \cdot \left(kp^{hp} + TF^{hp}\right)} - b_{tg} \cdot TG \quad (41)$$

Model 7: AND-gate: SMAD repressor, TF repressor.

$$\frac{d\text{TG}}{dt} = syn_{tg} \cdot \frac{kp^{hp} \cdot ks_1^{hs_1}}{\left(\text{SMAD}^{hs_1} + ks_1^{hs_1}\right) \cdot \left(kp^{hp} + TF^{hp}\right)} - b_{tg} \cdot TG \quad (42)$$

Model 8: OR-gate: SMAD repressor, TF repressor.

$$\frac{d\text{TG}}{dt} = syn_{tg} \cdot \left(\frac{ks_1^{hs_1}}{\text{SMAD}^{hs_1} + ks_1^{hs_1}} + \frac{kp^{hp}}{kp^{hp} + TF^{hp}}\right)$$
$$- syn_{tg} \cdot \frac{kp^{hp} \cdot ks_1^{hs_1}}{\left(\text{SMAD}^{hs_1} + ks_1^{hs_1}\right) \cdot \left(kp^{hp} + TF^{hp}\right)} - b_{tg} \cdot TG \quad (43)$$

Similar to the simple model, we calculated the steady state without TGF$\beta$ stimulation. For t = 0 and SMAD = 0, we get the following steady states per model:

Model 1: AND-gate: SMAD activator, TF activator.

$$\frac{syn_{tg}}{b0} \quad (44)$$

Model 2: OR-gate: SMAD activator, TF activator.

$$\frac{syn_{tg} + \frac{rp}{kp^{hp} + 1}}{b_{tg}} \quad (45)$$

Model 3: AND-gate: SMAD repressor, TF activator.

$$\frac{syn_{tg} + \frac{rp1}{kp^{hp} + 1}}{b_{tg}} \quad (46)$$

Model 4: OR-gate: SMAD repressor, TF activator.

$$\frac{syn_{tg}}{b_{tg}} \quad (47)$$

Model 5: AND-gate: SMAD activator, TF repressor.

$$\frac{syn}{b_{tg}} \quad (48)$$

Model 6: OR-gate: SMAD activator, TF repressor.

$$\frac{syn_{tg} \cdot kp^{hp}}{b_{tg} \cdot (kp^{hp} + 1)} \quad (49)$$

Model 7: AND-gate: SMAD repressor, TF repressor.

$$\frac{syn_{tg} \cdot kp^{hp}}{b_{tg} \cdot (kp^{hp} + 1)} \quad (50)$$

Model 8: OR-gate: SMAD repressor, TF repressor.

$$\frac{syn_{tg}}{b_{tg}} \quad (51)$$

The final equations used for FFL models (Fig 4D and E, Table S5) are shown in methods: *feedforward loops equation*.

## SMAD2 live-cell imaging upon TGF$\beta$-stimulated transcription factor KD cells

For live-cell time-lapse microscopy, cells were plated and transfected with the corresponding siRNA, as described in methods. For this approach, $1 \times 10^5$ cells in 1 ml cell culture medium lacking any antibiotics, 50 µl Opti-MEMTM (#1985062), 0.5 µl siRNA (stock concentration: 20 µM) and 1 µl RNAiMAX, were seeded in a polymer coverslip bottom 24 well plate (ibidi) 2 d before the experiment. Before starting the experiment, cells were washed once with 1 × PBS, and the medium was changed to DMEM lacking phenol red, supplemented with all growth factors, 5% horse serum, and antibiotics. To maintain constant $CO_2$ concentration (5%), temperature (37°C), and humidity, the microscope was placed in a custom incubator. Cells were imaged using a Nikon Ti inverted fluorescence microscope with a Nikon DS-Qi2 camera and a 20× plan apo objective (NA 0.75). The following filter sets were used: Venus (500/20 nm excitation, 515 nm dichroic beam splitter, 535/30 nm emission) and CFP (436/20 nm emission, 455 nm dichroic beam splitter, 480/40 nm excitation). Images were acquired every 10 min for the duration of the experiment using Nikon Elements software. TGF$\beta$-1 was prepared in 125 µl media and added, after two loops of images, to achieve the final concentration in 625 µl media. See Table S7 for measured cell numbers per condition.

## Gene-set-enrichment-analysis of dose-discriminating genes

The g:Profiler was used to analyze enriched genes among the dose-discriminating genes (Kolberg et al, 2023). For multiple testing correction, the build-in method g:SCS threshold was applied. We only considered KEGG and REACTOME gene sets and filtered for adjusted *P*-value < 0.01.

## Motif analysis

To investigate transcription factor binding in promoter regions, we first obtained the promoter sequences for FN1 and SERPINE1 genes using the Ensembl REST API. For each gene, a promoter region was defined as 2,000 base pairs (bp) upstream and 100 bp downstream of the transcription start site. The strand orientation, upstream, and downstream regions were adjusted accordingly. Promoter sequences were stored in FASTA format for subsequent analysis. This analysis was conducted in Python (version 3.12.3) using the library called request. To analyze the transcription factor binding potential in the retrieved promoter sequences, we performed motif scanning using the JUNB transcription factor Position Weight Matrix (PWM). The PWM for JUNB was retrieved from the JASPAR 2020 database (JASPAR ID: MA0490.2) via the JASPAR2020 package in R. The PWM was applied to the promoter sequences using the TFBSTools package, scanning for binding sites with a minimum score threshold of 80%. The promoter sequences, stored in FASTA files, were read using the Biostrings package in R. For each sequence, matches to the JUNB PWM were identified.

## Western blot analysis

For Western blot analysis, cells were seeded 2 d before stimulation with TGFβ. Upon stimulation, cells were harvested and lysed at the indicated time points, using RIPA buffer including protease and phosphatase inhibitors (Halt Protease and Phosphatase Inhibitor Cocktail, Thermo Fisher Scientific). The protein concentration was measured by Pierce BCA assay (Thermo Fisher Scientific) according to the manufacturer instruction. From each sample, 20 μg of protein was separated by electrophoresis, using Novex WedgeWell 10%, Tris-Glycin Mini-Gels (Thermo Fisher Scientific), and transferred to a PVDF Immobilion FL membrane (Sigma-Aldrich), using the iBlot system (Thermo Fisher Scientific). After protein transfer, the membrane was dried for 1 h and rehydrated for 1 min using 100% methanol and 1xTBS for 5 min. Membranes were blocked with Intercept Blocking solution (#927-60001; LiCorbio) for 1 h. To detect specific proteins of interest primary antibodies from Cell signaling (#3108, 1:1,000; pSMAD2), Santa Cruz (#sc-8051, 1:1,000; JUNB), and Proteintech (#68056-1-Ig, #11290-2-AP, 1:10,000; RPLP0) were diluted in the Intercept blocking buffer, containing 0.2% Tween20, and incubated overnight at 4°C. Fluorescent-labelled secondary antibodies (680RD Goat anti-Mouse IgG, # 926-68070; IRDye, 800CW Goat anti-Rabbit IgG, #926-32211; IRDye) were diluted in Intercept blocking buffer, containing 0.2% Tween20 and 0.02% SDS in which the membrane was incubated for 1 h at RT. Proteins were detected with a LiCor Odyssey M imager.

## Data Availability

RNaseq data and the code of mathematical modeling are publicly available via DOI: https://doi.org/10.5281/zenodo.10962767.

## Supplementary Information

## Acknowledgements

We thank Monilola Olayioye and Merih Özverin (Institute of Cell Biology and Immunology, University of Stuttgart) for the help with flow-cytometry analysis, the core facility at the IMB in Mainz for performing RNA sequencing and Clemens Kreutz for the discussion on model acceptance analysis. The authors acknowledge the support by the state of Baden-Württemberg through bwHPC. Portions of this work were developed from the doctoral dissertations of L Hartmann and P Kristofori. This work was supported by DFG funding to A Loewer (LO 1634/7-1) and S Legewie (LE 3473/4-1).

### Author Contributions

L Hartmann: conceptualization, data curation, formal analysis, validation, investigation, visualization, methodology, project administration, and writing—original draft, review, and editing.
P Kristofori: data curation, formal analysis, validation, investigation, visualization, and methodology.
C Li: data curation, formal analysis, validation, investigation, visualization, and methodology.
K Becker: formal analysis and methodology.
L Hexemer: formal analysis and methodology.
S Bohn: formal analysis and methodology.
S Lenhardt: data curation, formal analysis, investigation, and methodology.
S Weiss: data curation, formal analysis, and methodology.
B Voss: formal analysis and methodology.
A Loewer: conceptualization, supervision, funding acquisition, investigation, and project administration.
S Legewie: conceptualization, data curation, formal analysis, supervision, funding acquisition, validation, investigation, visualization, methodology, project administration, and writing—original draft, review, and editing.

### Conflict of Interest Statement

The authors declare that they have no conflict of interest.

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
