## [Reviewer comments · Life Science Alliance]

Life Science Alliance

Transcriptional regulators ensuring specific gene expression and decision making at high TGF β doses

Laura Hartmann, Panajot Kristofori, Congxin Li, Kolja Becker, Lorenz Hexemer, Stefan Bohn, Sonja Lenhardt, Sylvia Weiss, Björn Voss, Alexander Loewer, and Stefan Legewie

DOI: <https://doi.org/10.26508/lsa.202402859>

Corresponding author(s): Stefan Legewie, and Alexander Loewer, Technical University of Darmstadt

Review Timeline:

Submission Date:	2024-06-02
Editorial Decision:	2024-07-17
Revision Received:	2024-10-17
Editorial Decision:	2024-10-17
Revision Received:	2024-10-30
Accepted:	2024-10-31

Transaction Report:

July 17, 2024

Re: Life Science Alliance manuscript #LSA-2024-02859-T

Prof Stefan Legewie
University of Stuttgart
Institute for Biomedical Genetics, Systems Biology
Germany

Dear Dr. Legewie,

Thank you for submitting your manuscript entitled "Transcriptional regulators ensuring specific gene expression and decision making at high TGF β doses" to Life Science Alliance. The manuscript was assessed by expert reviewers, whose comments are appended to this letter. We invite you to submit a revised manuscript addressing the Reviewer comments.

Thank you for this interesting contribution to Life Science Alliance. We are looking forward to receiving your revised manuscript.

Sincerely,

B. MANUSCRIPT ORGANIZATION AND FORMATTING:

Reviewer #1 (Comments to the Authors (Required)):

In the manuscript „Transcriptional regulators ensuring specific gene expression and decision making at high TGFb doses“, the groups of Alexander Loewer and Stefan Legewie use time-resolved transcriptome analysis and defined perturbations combined with mathematical modelling to dissect the transcriptional response and transcription factor network architecture downstream of TGFb/SMAD signaling. The main result is that JUNB and SMAD work in a feed-forward loop architecture for a subset of SMAD responsive genes, particularly those that respond to strong TGFb stimulation and are involved in EMT. They nicely show that JUNB's role is to improve the discrimination of low and high doses of TGFb.

The paper is very well written and constitutes an innovative approach to the problem of signal decoding.

I have only minor comments that could improve the overall readability of the study.

- 1) on page 6, the order of the pattern does not follow the order that is in Figure 4A+B, and Figure 4A+B don't have the same order. It would be much easier to grasp if text and both subfigures would have the same order.
- 2) The later parts of the results are very technical in tone (but well done!). Within all the details, it may be difficult to still grasp the main message (i.e. the feed forward network and its function). For the overall understanding, it may be therefore worth considering adding a concept figure describing the network topology and how it relates to the main finding as a last figure, which is then referred to from the discussion.

Reviewer #2 (Comments to the Authors (Required)):

This study investigates the mechanisms of decision making towards specific cellular fates in response to TGFβ. By combining SMAD transcription factor imaging, RNA sequencing, and morphological assays, the authors identified repressors and activators of TGFβ-signaling. The authors presented a global view of TGFβ-dependent gene regulation and described specific mechanisms involved in TGFβ-signaling and decision making. Overall, the study is interesting and promising. However, the authors should address the following comments.

1. The authors should add a western blot analysis for the nuclear and cytoplasmic fractions of SMAD2 to support of the live-cell imaging results.
2. The authors should show a time-course (Western blot analysis) to confirm that JUNB is a late activator.
3. To confirm the results that only higher doses of TGFβ significantly induce EMT in MCF10A cells, the authors should treat MCF10A cells with a range of TGFβ concentrations and assess the expression of EMT markers at various time points. This would help establish the threshold for EMT induction.
4. The authors should perform functional assays to evaluate EMT-associated cellular behaviors such as migration assays or invasion assays.
5. The authors should show the phosphorylation of SMAD proteins by western blot during the induction of EMT. Confirming the dynamics of SMAD phosphorylation would strengthen the paper.
6. Is there any binding site predicted for JUNB on the promoters of SERPINE1 and FN1? this could provide more direct evidence of JUNB's role in regulating these genes.

Reviewer #3 (Comments to the Authors (Required)):

The manuscript by Hartmann et al explores the transcriptional changes induced by TGFb-treatment of MCF10A cells in response to two different doses of ligand treatment (25pM and 100pM) over time via RNAseq. The team then investigates the roles of potential downstream transcriptional regulators using siRNA and monitoring the responses to TGFb ligand stimulation again via RNAseq. Mathematical modeling is used to try and understand the responses to ligand treatment.

The study appears well executed, with controls well explained. The data could help understanding of TGFB signaling dynamics

Some limitations of the study include:

It is not clear how many RNAseq replicates were done at each timepoint, nor whether the data are being made available to the public.

Given the nature of the single methodology type used in the study, there is a limitation to how much can be concluded just with RNAseq. For example, the authors try and infer RNA stability in their analysis. At a minimum, the authors should try and compare intronic vs. exonic RNA to try and infer RNA stability levels vs. de novo transcription rates.

It is surprising that all of the negative regulators induced upon TGFB treatment have an effect broadly on TGFB target genes. Shouldn't there be redundancy between factors working in a similar capacity? Or are all these factors working together in a complex?

All of the experiments are done in a single cell line - the authors should indicate the potential limitations of the findings and how they could apply to other tissue types.

Only loss of function experiments are performed for the downstream transcription factors (siRNA). It would be a more rigorous study if gain-of-function studies complemented these efforts. How would overexpression of JUN for example, affect the transcriptional dynamics in the cell?

We thank all reviewers for evaluating our manuscript, for their positive feedback and constructive criticism, which helped us to substantially improve the paper. As detailed in the point-by-point response below, we included the following major changes to the manuscript:

- **changed the order of rejected gene groups in Figure 4 A+B (Reviewer 1)**
- **included a conceptual summarizing figure showing how dose-discriminating genes are regulated by co-repressors and co-activators through FFLs (Reviewer 1)**
- **measured SMAD2 phosphorylation und JUNB protein expression dynamics (Reviewer 2)**
- **determined the threshold of EMT induction (Reviewer 2)**
- **included a motif analysis for JUNB binding in FN1 and SERPINE1 promoter regions (Reviewer 2)**
- **discussed JUNB-dependent TGF β -induced EMT in other cell lines (Reviewer 3)**
- **provided our data sharing plan (Reviewer 3)**
- **discussed the action of repressors affecting a broad range of target genes (Reviewer 3)**
- **discussed how JUNB overexpression promotes EMT in MCF10A cells (Reviewer 3)**

Reviewer #1 (Comments to the Authors (Required)):

In the manuscript „Transcriptional regulators ensuring specific gene expression and decision making at high TGF β doses“, the groups of Alexander Loewer and Stefan Legewie use time-resolved transcriptome analysis and defined perturbations combined with mathematical modelling to dissect the transcriptional response and transcription factor network architecture downstream of TGF β /SMAD signaling. The main result is that JUNB and SMAD work in a feed-forward loop architecture for a subset of SMAD responsive genes, particularly those that respond to strong TGF β stimulation and are involved in EMT. They nicely show that JUNB's role is to improve the discrimination of low and high doses of TGF β . The paper is very well written and constitutes an innovative approach to the problem of signal decoding.

I have only minor comments that could improve the overall readability of the study.

1. On page 6, the order of the pattern does not follow the order that is in Figure 4A+B, and Figure 4A+B don't have the same order. It would be much easier to grasp if text and both subfigures would have the same order.

We addressed this issue and present the kinetics of the rejected genes in both the main text and Figure 4 A+B in the same order: delayed, inconsistent, biphasic, continuous, immediate early. Please see the revised version of Figure 4.

2. The later parts of the results are very technical in tone (but well done!). Within all the details, it may be difficult to still grasp the main message (i.e. the feed forward network and its function). For the overall understanding, it may be therefore worth considering adding a concept figure describing the network topology and how it relates to the main finding as a last figure, which is then referred to from the discussion.

As suggested by the reviewer, we included a conceptual figure as final Figure 7 summarizing the most important findings and conclusions of our work, and refer to it in the revised discussion. The figure is also shown below (Figure 1).

Figure 1: The combination of multiple SMAD-dependent feedforward loops establishes dose discrimination at the level of target gene expression. SMAD TFs control expression of DDGs directly and indirectly by inducing the expression of transcriptional repressors (e.g., SKIL) and activators (e.g., JUNB). The repressors prevent the early upregulation of DDGs, and the activators lead to their late amplification at the high TGFβ dose, thereby leading to dose-discrimination.

Reviewer #2 (Comments to the Authors (Required)):

This study investigates the mechanisms of decision making towards specific cellular fates in response to TGF β . By combining SMAD transcription factor imaging, RNA sequencing, and morphological assays, the authors identified repressors and activators of TGF β -signaling. The authors presented a global view of TGF β -dependent gene regulation and described specific mechanisms involved in TGF β -signaling and decision making. Overall, the study is interesting and promising. However, the authors should address the following comments.

1. The authors should add a western blot analysis for the nuclear and cytoplasmic fractions of SMAD2 to support of the live-cell imaging results.

We agree with the reviewer that it is important to validate the nuclear translocation dynamics live-cell SMAD2 reporter. In our previous work, we performed anti-SMAD2 immunofluorescence (IF) to validate that the localization of the SMAD2 reporter reflects the dynamics of endogenous SMAD2 in MCF10A cells (Fig. EV1H in Strasen *et al.*, 2018). Specifically, we correlated the nuclear IF signal of total SMAD2 (endogenous plus reporter) to the YFP intensity in the nucleus, and found an excellent correlation with Pearson correlation coefficient of 0.88, even though the SMAD2 reporter is expressed at low levels relative to the endogenous protein (Fig. EV1 H). This validates the translocation of the reporter can be confirmed by orthogonal methods. Given the peculiarities of nucleocytoplasmic fractionation, we think that immunofluorescence experiments mentioned above already present a better validation of the live-cell reporter system, and therefore did not do the suggested fractionation experiment but refer to the published results on page 4 of the revised manuscript.

2. The authors should show a time-course (Western blot analysis) to confirm that JUNB is a late activator.

As suggested by the reviewer, we performed time-resolved JUNB western blotting. However, using two distinct antibodies, we could either not detect an upregulation (Cell signaling, #3753) (Figure 2, A), or found only a very weak upregulation (Santa Cruz, #sc-8051) at the protein level, when treated with 100 pM TGF β (Figure 2, B, E).

Previous studies analyzing TGF β -induced JUNB protein upregulation in MCF10A cells reported either a strong upregulation (Sundqvist *et al.*, 2018), or found only a very modest induction of JUNB levels as we report here (Antón-García *et al.*, 2023). The main reason for the discrepancy between labs may be the culture conditions: when we treated serum-starved cells with varying TGF β doses, we found stronger induction of JUNB by the high dose (100 pM TGF β), whereas the protein was unchanged in response to the low dose (2.5 pM) (Figure 2, C). Thus, the growth factor stimulation status before TGF β treatment may determine whether JUNB is upregulated or not.

Reasoning that JUNB may be regulated by phosphorylation, not transcriptional induction, we also performed time-resolved western blotting using a phospho-specific antibody recognizing phosphorylation at Thr102 and Thr104 (Cell signaling, #8053), which known to be targeted by Jnk signaling, a non-canonical pathway controlled by TGF β . Even though Anisomycin as a positive control stimulus induced strong JUNB phosphorylation in

MCF10A cells, we could not detect a reliable signal in response to TGF β treatment at various time points (1.5, 3, 6, 12, 24 h) (Figure 2, D). This excludes that TGF β regulates JUNB by means of phosphorylation at sites Thr102 and Thr104.

Taken together, our data in which JUNB siRNA dampens late gene expression amplification, but JUNB protein is only weakly induced by TGF β , suggests that JUNB is recruited indirectly to target gene promoters, e.g. by a cooperating transcription factor, whose activity is controlled by TGF β . According to the literature, JUNB indeed cooperates with many TFs including other Jun family members, the Fos, ATF, and MAF family members (Bejjani *et al.*, 2019), but also SMAD3 and SMAD4 (Liberati *et al.*, 1999).

We felt that following this up further would be beyond the scope of this paper and therefore included the above-mentioned result in the Supplement (Supplementary Figure 10), referring to in the results and discussion sections of the revised manuscript

Figure 2: Western blot analysis of JUNB protein expression and phosphorylation in MCF10A cells upon TGF β stimulation. **A)** JUNB expression levels in MCF10A cells treated with TGF β (2.5 pM and 100 pM) over various time points (0, 0.75, 1.5, 3, 6, 12, 24 h), detected using JUNB antibodies from Cell Signaling. **B)** JUNB expression levels in MCF10A cells treated with high dose TGF β (100 pM) at the indicated time points (0, 1.5, 3, 6, 12, 24 h), detected using JUNB antibodies from Santa Cruz. **C)** JUNB expression in serum-starved MCF10A cells treated with low and high dose TGF β (2.5 pM and 100 pM), using JUNB antibodies from Cell Signaling. **D)** Phosphorylated JUNB (pJUNB) levels in MCF10A cells stimulated with high dose TGF β (100 pM) or treated with 25 μ g/ml Anisomycin for 15 min, 30 min or 120 min as a positive control. RPLP0 was used as the loading control in all experiments. **E)** Time-resolved quantification of JUNB levels after stimulation with 100 pM TGF β , assessed by Western blot analysis across four different replicates using JUNB antibodies from Santa Cruz. JUNB expression was normalized to the housekeeping protein

RPLP0 at each time point and then to the control (t_0) to calculate fold changes (FC). Mean values and SEM of the normalized *JUNB* expression is shown.

3. To confirm the results that only higher doses of TGF β significantly induce EMT in MCF10A cells, the authors should treat MCF10A cells with a range of TGF β concentrations and assess the expression of EMT markers at various time points. This would help establish the threshold for EMT induction.

In the original manuscript, we measured the EMT markers (Vim, E-Cadh) at different doses upon stimulation for 5d (Figure 2, D), and at different time points for 100 pM TGF β (Supplementary Figure 8, C). The dose-dependent data showed that EMT is not or only weakly induced by low doses of TGF β (1 pM, 5 pM), and that a saturating stimulus (250 pM) is required for maximal induction 5d post stimulation. In the 100 pM time course, EMT induction was more pronounced 8d when compared to 5d post-stimulation for a saturating TGF β stimulus (100 pM). Hence, as commented by the reviewer, more measurements at various time points and TGF β doses would be required to determine the threshold of EMT induction, as it might be possible that EMT markers are upregulated 8d post stimulation also for the lower doses. In this case, EMT induction would be merely delayed but still proceed at low doses.

To exclude the possibility of late EMT induction at the 2.5 pM dose, we performed an 11d stimulation experiment at low, intermediate, and high doses and analyzed the cell morphology using light microscopy, reasoning that epithelial cells show apical-basal cell-polarity and cell-cell adhesion, whereas mesenchymal cells exhibit an elongated spindle shape. In ctrl and 2.5 pM TGF β stimulation conditions cells maintain their cell polarity and cell-cell adhesion over the complete 11d time course, indicating the absence of EMT. In contrast, for 100 pM TGF β cells became increasingly spindle-shaped at 5d, 8d, and especially 11d post stimulation. At an intermediate stimulus (5 pM), spindle-shaped cells were transiently visible at 5d and 8d, but not at 11d. We conclude that the low 2.5 pM TGF β dose was insufficient to induce EMT also for long-term stimulation, indicating that a concentration of 5 pM TGF β marks the threshold for transient EMT induction, and that saturating TGF β was required for sustained EMT. These data agree well with our finding, that DDGs controlling EMT do not show late amplification at 2.5 pM. We include this new data in the revised manuscript (Supplementary Figure 1, B), and refer to it on page 5 of the revised manuscript.

B

Morphological change of TGF β stimulated MCF10A cells

Figure 3: Morphological changes of TGF β stimulated MCF10A cells upon long term treatment (2d, 5d, 8d, 11d) with different TGF β doses (2.5, 5, and 100 pM) shown by wide-field microscopy of the cell culture. Low TGF β dose (2.5 pM) show cell morphology similar to that of control cells, maintaining apical-basal polarity over the complete 11d time course. High TGF β dose (100pM) changes the cell morphology, leading to the loss of apical-basal polarity and a spindle-shaped appearance at 5d, 8d, and especially 11d post-stimulation. Intermediate TGF β dose (5pM) show spindle-shaped MCF10A cells only 5d and 8d, but not 11d post-stimulation. Scale bar with 200 μ m is indicated in the top left.

4. The authors should perform functional assays to evaluate EMT-associated cellular behaviors such as migration assays or invasion assays.

In our previous work, we simultaneously performed a quantification of cell migration in the live-cell imaging data, did scratch assays and EMT marker expression at multiple conditions (i.e., application of different TGF β ligands in proliferating and quiescent cells) to confirm that altered EMT marker expression is accompanied by increased cell migration. We refer to this work in the revised manuscript (Strasen *et al.*, 2018; Bohn *et al.*, 2023). In the context of our EMT analysis in response to JUNB KD, we additionally cite work by another group that showed reduced TGF β -induced invasive behavior upon JUNB KD by

growing MCF10A cells as three-dimensional spheroids embedded in collagen I (Antón-García *et al.*, 2023).

5. The authors should show the phosphorylation of SMAD proteins by western blot during the induction of EMT. Confirming the dynamics of SMAD phosphorylation would strengthen the paper.

We agree with the reviewer that the dynamics of SMAD phosphorylation are an important measure of pathway activation and likely a key determinant of EMT. In our previous work, we had measured SMAD phosphorylation for up to 6h after TGF β stimulation by Western Blot and found that the temporal dynamics of the SMAD2 live-cell reporter quantitatively agrees with the SMAD2 phosphorylation time course (Fig. 1C and EV1A in Strasen *et al.*, 2018). Furthermore, performed time course IF experiments using an anti-phospho SMAD2 antibody over 8h post TGF β stimulation and confirmed that the SMAD2 phosphorylation dynamics in parental MCF10A cells shows a quantitative agreement with the nucleocytoplasmic ratio determined by the SMAD2 imaging approach (Fig. EV1I in Strasen *et al.*, 2018). This data indicates that the SMAD2 reporter quantitatively reflects SMAD2 phosphorylation.

To connect the SMAD2 signaling dynamics to the initiation of EMT, we performed a long-term 5d live-cell imaging time course and reported continuous SMAD2 activation for very high TGF β doses (250 pM), while at low doses (2.5 pM TGF β) the signaling terminated early and was restored when replenishing the ligand (Fig. 3A in Bohn *et al.*, 2023). Similar high vs. low dose dynamics were reported for SMAD2 phosphorylation in the literature (Clarke *et al.*, 2009; Zi *et al.*, 2011). Interestingly, the total integrated signal of the live-cell reporter (area-under-curve, AUC), and thus of SMAD2 phosphorylation, was predictive for cell motility, cell proliferation, and the magnitude of global gene expression (Fig 3 D-F and Fig. 5F in Bohn *et al.*, 2023). Therefore, the SMAD2 signal indeed predicts EMT initiation.

To further address the reviewer comment, we performed the suggested additional Western Blot experiment and followed SMAD2 phosphorylation for 24h after low- or high-dose TGF β stimulation. Again, the observed dynamics were highly similar to the dynamics of SMAD2 nuclear translocation, with a transient peak at 2.5 pM TGF β and a sustained signal at 100 pM TGF β . This argues that the conclusions derived from the live-cell reporter reflect SMAD2 phosphorylation, and that both dynamics are related to the initiation of EMT.

In the revised manuscript, we discuss on pages 4 and 5 that the dynamics measured the live-cell imaging reporter reflect the time course of endogenous SMAD2 phosphorylation, and that sustained SMAD2 phosphorylation is required to induce EMT. We include time resolved SMAD2 phosphorylation data upon low and high dose TGF β in the revised manuscript (Supplementary Figure 1, A).

A

Figure 4: Time-resolved analysis of pSMAD2 levels after stimulation with 2.5 pM (blue) and 100 pM (orange) TGFβ, assessed by Western blot analysis across four different replicates. pSMAD2 signals were normalized to the housekeeping protein RPLP0 at each time point. Mean values and SEM of the normalized pSMAD2 signals are shown (n=4). For t=0, error bars are not visible due to very small SEMs at this time point.

6. Is there any binding site predicted for JUNB on the promoters of SERPINE1 and FN1? this could provide more direct evidence of JUNB's role in regulating these genes.

We performed a motif search to determine whether JUNB binds to the promoters of SERPINE1 or FN1. Our analysis indicates that JUNB has 3 binding sites in the FN1 promoter and 4 in the SERPINE1 promoters. The specific JUNB binding sites to FN1 and SERPINE1 promoters are shown in the Supplementary Table S8 of the revised manuscript. The method description of this motif analysis is included in the Supplementary text. In addition, to validate the motif analysis, we found published JUNB ChIP-seq data recorded in A375 cells upon TGFβ stimulation (GSE213326).

Figure 5: JUNB ChIPseq tracks of FN1 and SERPINE1 in A375 cells

In line with the motif search, there are ChIP-seq peaks indicating strong JUNB binding to FN1 and SERPINE1 (Figure 5). Therefore, the evidence supports JUNB's regulatory role as a co-factor in the feed-forward loop.

Reviewer #3 (Comments to the Authors (Required)):

The manuscript by Hartmann et al explores the transcriptional changes induced by TGF β -treatment of MCF10A cells in response to two different doses of ligand treatment (25pM and 100pM) over time via RNAseq. The team then investigates the roles of potential downstream transcriptional regulators using siRNA and monitoring the responses to TGF β ligand stimulation again via RNAseq. Mathematical modeling is used to try and understand the responses to ligand treatment.

The study appears well executed, with controls well explained. The data could help understanding of TGF β signaling dynamics

Some limitations of the study include:

1. It is not clear how many RNAseq replicates were done at each timepoint, nor whether the data are being made available to the public.

We provided this information in the corresponding methods section. For each RNAseq data set, three biological replicates were conducted. The RNAseq data and MATLAB scripts of the mathematical model will be publicly available under DOI: 10.5281/zenodo.10962767.

2. Given the nature of the single methodology type used in the study, there is a limitation to how much can be concluded just with RNAseq. For example, the authors try and infer RNA stability in their analysis. At a minimum, the authors should try and compare intronic vs. exonic RNA to try and infer RNA stability levels vs. de novo transcription rates.

We agree with the reviewer that inferring mRNA stability without orthogonal experiments can be difficult and quantifying pre-mRNA maturation process will facilitate faithful estimation of mRNA half-lives. Unfortunately, we performed polyA-enrichment in our RNA sequencing experiments, which typically isolates mature mRNA and lacks reliable representation of pre-mRNAs, therefore practically hindering the requested analysis.

To test the inferred RNA stability in an alternative way, we directly correlated our model-fitted mRNA half-lives to a compendium of publicly available mRNA half-life measurements (Agarwal and Kelley, 2022). We obtained mild correlations when individually correlating the model-inferred half-lives to 36 datasets from adherent cell lines, with median and maximum Spearman correlation coefficients of 0.19 and 0.29, respectively (Figure 6). However, the measured RNA half-lives already showed comparable variation across different labs using similar pulse-labeling methods: Even though datasets obtained within the same lab show quite high consistency (median correlation 0.9), the half-lives measured by different groups are much less correlated, and the median data-data correlation coefficient across research groups is 0.2 (Figure 7), which is close to the median model-data correlation (0.19). This suggests that the uncertainty in our inferred mRNA half-lives is comparable to the uncertainty of half-life measurements across research groups and cell lines.

Further experiments, such as the ones the reviewer suggested, need to be performed to precisely determine the mRNA stability. However, we believe that this is not the focus of the current manuscript and warrants an independent study, as we focused on inferring

direct targets of SMAD signaling and did not focus on an analysis of model-inferred half-lives.

Figure 6: Comparison of mRNA half-lives from model inference to measurements from public databases (Agarwal and Kelley, 2022). Spearman's correlation was calculated between the model and 36 datasets from adherent cell lines.

Figure 7: Moderate correlation of model-inferred mRNA half-lives to publicly available half-life measurements matches measurement-to measurement correlation across laboratories. Spearman's correlation coefficients within the 36 RNA half-life datasets were calculated for datasets obtained within the same research group (gray), across research groups (blue) and compared to the model correlation to all datasets (red). The vertical lines show the median values of the three groups.

3. It is surprising that all of the negative regulators induced upon TGF β treatment have an effect broadly on TGF β target genes. Shouldn't there be redundancy between factors working in a similar capacity? Or are all these factors working together in a complex?

We agree with the reviewer that the observed broad effects of negative regulators are surprising, but would like to point out that the repressive effects were reproducible across three biological RNAseq replicates, and could be confirmed in independent qPCR experiments (at least for the tested genes and conditions). In our opinion, the most plausible mode of action of repressors is direct binding to SMAD proteins which has been reported for SKI (Luo *et al.*, 1999), SKIL (Stroschein *et al.*, 1999), SNAI1 (Vincent *et al.*, 2009), and RUNX1 (Hanai *et al.*, 1999). Hence, the repressors do not necessarily function in the same complex, but they all modulate SMAD proteins as the common denominators, which would explain their global effects on TGF β -dependent gene expression in MCF10A cells. Our data does not rule out redundancy, since it is possible that combined knockdowns would have much stronger effects on TGF β -dependent gene regulation when compared to the so-far-tested individual depletions.

We briefly mention in the revised Discussion that complex formation with the SMADs may explain global and overlapping repressor effects as follows (p. 12): "It is surprising that all of the negative regulators induced upon TGF β treatment have a broad effect on TGF β target genes. One possible explanation may be that many of them exert their effects by direct binding to SMAD proteins (Hanai *et al.*, 1999; Luo *et al.*, 1999; Stroschein *et al.*, 1999; Vincent *et al.*, 2009), acting redundantly or complementary by modulating the recruitment of epigenetic co-repressors or co-activators to SMAD complexes."

4. All of the experiments are done in a single cell line - the authors should indicate the potential limitations of the findings and how they could apply to other tissue types.

The reviewer correctly points out that our findings are currently limited to the MCF10A cell line. However, similar dose-dependent SMAD signaling dynamics were measured in other cell lines, e.g., in HaCaT or PE25 cells (Zi et al., 2011, Clarke et al., 2009). Therefore, it is plausible that the feedforward loops we discuss here are similarly involved in signal decoding, and EMT regulation in these cell types.

Accordingly, JUNB-dependent TGF β -induced EMT was reported in mouse mammary epithelial NMuMG cell line and the mouse kidney epithelial MCT cell line (Gervasi et al., 2012). Moreover, xenograft models of Gall Bladder Cancer (GBC) cells showed that JUNB induces cell migration and invasion (Lian et al., 2015), and a study of U2OS cells indicated that JUNB promotes tumor development and metastasis (Pérez-Benavente et al., 2022). However, cell type specificity certainly occurs, as JUNB did not show EMT-regulatory capacity in A549 lung cancer cells (Antón-García et al., 2023). We discuss these issues in the revised Discussion (p. 13).

5. Only loss of function experiments are performed for the downstream transcription factors (siRNA). It would be a more rigorous study if gain-of-function studies complemented these efforts. How would overexpression of JUN for example, affect the transcriptional dynamics in the cell?

We agree with the reviewer that gain-of-function experiments are potentially an interesting avenue of research. However, previous research in MCF10A cells indicates that JUNB overexpression induces EMT in the absence of TGF β in MCF10A cells (Antón-García et al., 2023). Therefore, we do not believe that a JUNB overexpression experiment, with many shifts in basal gene expression, is a good setting to assess effects on transcriptional dynamics as opposed to the already existing KD experiment, where basal effects of JUNB siRNA before TGF β stimulation are negligible. We further felt that a detailed titration of the JUNB overexpression level followed by an assessment of global gene expression shifts by RNAseq is beyond the scope of the current paper. Therefore, we did not address this reviewer comment further, but appreciate the suggestion and cite the JUNB overexpression work in the revised Discussion.

References

- Agarwal, V. and Kelley, D. R. (2022) 'The genetic and biochemical determinants of mRNA degradation rates in mammals', *Genome Biology*, 23(1), p. 245. doi: 10.1186/s13059-022-02811-x.
- Antón-García, P. et al. (2023) 'TGF β 1-Induced EMT in the MCF10A Mammary Epithelial Cell Line Model Is Executed Independently of SNAIL1 and ZEB1 but Relies on JUNB-Coordinated Transcriptional Regulation.', *Cancers*, 15(2). doi: 10.3390/cancers15020558.
- Bejjani, F. et al. (2019) 'The AP-1 transcriptional complex: Local switch or remote command?', *Biochimica et biophysica acta. Reviews on cancer*, 1872(1), pp. 11–23. doi: 10.1016/j.bbcan.2019.04.003.

- Bohn, S. *et al.* (2023) 'State- and stimulus-specific dynamics of SMAD signaling determine fate decisions in individual cells', *Proceedings of the National Academy of Sciences*, 120(10), p. e2210891120. doi: 10.1073/pnas.2210891120.
- Clarke, D. C. *et al.* (2009) 'Transforming growth factor beta depletion is the primary determinant of Smad signaling kinetics.', *Molecular and cellular biology*, 29(9), pp. 2443–2455. doi: 10.1128/MCB.01443-08.
- Gervasi, M. *et al.* (2012) 'JunB contributes to Id2 repression and the epithelial-mesenchymal transition in response to transforming growth factor- β ', *Journal of Cell Biology*, 196(5), pp. 589–603. doi: 10.1083/jcb.201109045.
- Hanai, J. *et al.* (1999) 'Interaction and functional cooperation of PEBP2/CBF with Smads. Synergistic induction of the immunoglobulin germline Calpha promoter.', *The Journal of biological chemistry*, 274(44), pp. 31577–31582. doi: 10.1074/jbc.274.44.31577.
- Lian, S. *et al.* (2015) 'PDK1 induces JunB, EMT, cell migration and invasion in human gallbladder cancer.', *Oncotarget*, 6(30), pp. 29076–29086. doi: 10.18632/oncotarget.4931.
- Liberati, N. T. *et al.* (1999) 'Smads bind directly to the Jun family of AP-1 transcription factors.', *Proceedings of the National Academy of Sciences of the United States of America*, 96(9), pp. 4844–4849. doi: 10.1073/pnas.96.9.4844.
- Luo, K. *et al.* (1999) 'The Ski oncoprotein interacts with the Smad proteins to repress TGF β signaling.', *Genes & development*, 13(17), pp. 2196–2206. doi: 10.1101/gad.13.17.2196.
- Pérez-Benavente, B. *et al.* (2022) 'New roles for AP-1/JUNB in cell cycle control and tumorigenic cell invasion via regulation of cyclin E1 and TGF- β 2', *Genome Biology*, 23(1), p. 252. doi: 10.1186/s13059-022-02800-0.
- Strasen, J. *et al.* (2018) 'Cell-specific responses to the cytokine TGF β are determined by variability in protein levels', *Molecular Systems Biology*, 14(1), pp. 1–17. doi: 10.15252/msb.20177733.
- Stroschein, S. L. *et al.* (1999) 'Negative feedback regulation of TGF-beta signaling by the SnoN oncoprotein.', *Science (New York, N.Y.)*, 286(5440), pp. 771–774. doi: 10.1126/science.286.5440.771.
- Sundqvist, A. *et al.* (2018) 'JUNB governs a feed-forward network of TGF β signaling that aggravates breast cancer invasion', *Nucleic Acids Research*, 46(3), pp. 1180–1195. doi: 10.1093/nar/gkx1190.
- Vincent, T. *et al.* (2009) 'A SNAIL1-SMAD3/4 transcriptional repressor complex promotes TGF-beta mediated epithelial-mesenchymal transition.', *Nature cell biology*, 11(8), pp. 943–950. doi: 10.1038/ncb1905.
- Zi, Z. *et al.* (2011) 'Quantitative analysis of transient and sustained transforming growth factor- β signaling dynamics.', *Molecular systems biology*, 7, p. 492. doi: 10.1038/msb.2011.22.

October 17, 2024

RE: Life Science Alliance Manuscript #LSA-2024-02859-TR

Stefan Legewie

Dear Dr. Legewie,

Thank you for submitting your revised manuscript entitled "Transcriptional regulators ensuring specific gene expression and decision making at high TGF β doses". We would be happy to publish your paper in Life Science Alliance pending final revisions necessary to meet our formatting guidelines.

- please be sure that the authorship listing and order is correct
- please incorporate your supplemental material file into your main manuscript text
- please add ORCID ID for secondary corresponding author-they should have received instructions on how to do so
- please add the Twitter handle of your host institute/organization as well as your own or/and one of the authors in our system
- please use the [10 author names, et al.] format in your references (i.e. limit the author names to the first 10)
- please add your supplementary figure legends to the main manuscript text

A. FINAL FILES:

B. MANUSCRIPT ORGANIZATION AND FORMATTING:

****It is Life Science Alliance policy that if requested, original data images must be made available to the editors. Failure to provide**

original images upon request will result in unavoidable delays in publication. Please ensure that you have access to all original data images prior to final submission.**

The license to publish form must be signed before your manuscript can be sent to production. A link to the electronic license to publish form will be available to the corresponding author only. Please take a moment to check your funder requirements.

Sincerely,

October 31, 2024

RE: Life Science Alliance Manuscript #LSA-2024-02859-TRR

Prof. Stefan Legewie

Dear Dr. Legewie,

Thank you for submitting your Research Article entitled "Transcriptional regulators ensuring specific gene expression and decision making at high TGF β doses". It is a pleasure to let you know that your manuscript is now accepted for publication in Life Science Alliance. Congratulations on this interesting work.

DISTRIBUTION OF MATERIALS:

Again, congratulations on a very nice paper. I hope you found the review process to be constructive and are pleased with how the manuscript was handled editorially. We look forward to future exciting submissions from your lab.

Sincerely,
